# Microinvasion by *Streptococcus pneumoniae* induces epithelial innate immunity during colonisation at the human mucosal surface

Caroline M. Weight [1], Cristina Venturini[1], Sherin Pojar [2], Simon P. Jochems [2], Jesús Reiné[2], Elissavet Nikolaou[2], Carla Solórzano [2], Mahdad Noursadeghi [1], Jeremy S. Brown [3], Daniela M. Ferreira [2] & Robert S. Heyderman [1]

Control of *Streptococcus pneumoniae* colonisation at human mucosal surfaces is critical to reducing the burden of pneumonia and invasive pneumococcal disease, interrupting transmission, and achieving herd protection. Here, we use an experimental human pneumococcal carriage model (EHPC) to show that *S. pneumoniae* colonisation is associated with epithelial surface adherence, micro-colony formation and invasion, without overt disease. Interactions between different strains and the epithelium shaped the host transcriptomic response in vitro. Using epithelial modules from a human epithelial cell model that recapitulates our in vivo findings, comprising of innate signalling and regulatory pathways, inflammatory mediators, cellular metabolism and stress response genes, we find that inflammation in the EHPC model is most prominent around the time of bacterial clearance. Our results indicate that, rather than being confined to the epithelial surface and the overlying mucus layer, the pneumococcus undergoes micro-invasion of the epithelium that enhances inflammatory and innate immune responses associated with clearance.

[1] Division of Infection and Immunity, University College London, London, UK. [2] Department of Clinical Sciences, Liverpool School of Tropical Medicine, Liverpool, UK. [3] Department of Respiratory Medicine, University College London, London, UK. Correspondence and requests for materials should be addressed to C.M.W. (email: c.weight@ucl.ac.uk)

Colonisation of upper respiratory tract (URT) mucosa by a range of pathogenic bacteria is a necessary precursor to both disease and onward transmission. Although *Streptococcus pneumoniae* is a common coloniser of the human nasopharynx, it is estimated to be responsible for >500,000 deaths due to pneumonia, meningitis and sepsis in children under 5 years of age worldwide[1].

In Europe and North America, there has been a dramatic effect of pneumococcal conjugate vaccine (PCV) on vaccine serotype (VT) invasive disease, carriage and transmission[2]. However, the emergence of non-VT pneumococcal disease worldwide and the more modest impact of PCV on colonisation in high transmission settings, threaten this success[3]. Control of pneumococcal URT colonisation in humans is not fully understood[4], and so defining the mechanistic basis for host control of pneumococcal colonisation at the mucosal surface is therefore crucial for the further optimisation of therapeutic interventions which target carriage and transmission. The process of transmission is not fully understood, but mucosal inflammation, potentially enhanced by co-infection with viruses such as Influenza A, has been proposed to mediate bacterial shedding from the nasopharynx[5].

Naturally acquired immunity to *S. pneumoniae* proteins are primarily mediated by mucosal T cells and anti-protein antibodies, controlled by Treg[6–8]. The role of anti-capsule polysaccharide antibodies in naturally acquired immune control remains unresolved[9]. URT epithelium is central to this immunity[10], binding and transporting antibodies, sensing bacteria via a range of surface and intracellular pathogen-associated molecular patterns (PAMPs) receptors, and rapidly transducing signals to recruit innate and inflammatory immune mechanisms[11,12].

Murine models suggest that adherence of *S. pneumoniae* to the mucosal epithelium may be followed by paracellular transmigration and tight junction modulation[13,14]. In contrast, studies of immortalised epithelial cell monolayers implicate endocytosis of *S. pneumoniae*, mediated though pneumococcal protein C–polymeric immunoglobulin receptor interactions[15,16]. The relative importance of epithelial endocytosis and paracellular migration in colonisation and invasion remains uncertain[16], but could influence epithelial sensing of this otherwise extracellular pathogen[14]. For example this may include Nod1 signalling via peptidoglycan[17] and TLR4 signalling via pneumolysin, a pore-forming toxin that induces inflammation and mediates both clearance and transmission in an infant mouse model[18–20].

Much of what we understand of the control of pneumococcal colonisation is derived from epidemiological studies and murine carriage models. The experimental human pneumococcal carriage (EHPC) model provides a well-controlled, reproducible tool to characterise the cellular and molecular mechanisms that underlie pneumococcal colonisation in humans[21]. The model has revealed that pneumococcal carriage results in neutrophil degranulation and monocyte recruitment to the nasopharynx[22], protecting against re-challenge with the same strain for up to 1 year[21,23].

Here, we explore the hypothesis that pneumococcal–epithelial engagement dictates the inflammatory/innate immune response and therefore the outcome of colonisation. We use the EHPC model and human epithelial cell lines with different strains of *S. pneumoniae* to probe these mechanisms and derive an epithelial transcriptome module from the in vitro culture system to interrogate the host response in the EHPC model. We show that pneumococcal colonisation in humans is characterised by micro-colony formation, junctional protein association and migration across the epithelial barrier without disease, which we have termed micro-invasion. This pattern of bacterial-host cell association shapes epithelial sensing of *S. pneumoniae*, which is partially pneumolysin dependent. Together, our data suggest that pneumococcal engagement with the epithelium in the early phases of colonisation may occur without eliciting a marked host response, but as colonisation becomes more established, epithelial sensing of *S. pneumoniae* enhances innate immunity/inflammation which we propose promotes clearance.

## Results

**Adhesion, micro-colony formation and micro-invasion in EHPC model.** We have used an EHPC model[21] to capture the epithelial events central to pneumococcal colonisation, obtaining human mucosal samples by curette biopsy. We defined a carrier as an individual colonised by 6B *S. pneumoniae* as detected by culture of nasal wash at any day following inoculation. Clearance was defined as when *S. pneumoniae* colonisation became negative by culture of nasal wash.

Colonisation was detected in 9/13 healthy volunteers by culture, 12/13 by microscopy (using 6B capsule-specific antibody labelling) and 9/11 by LytA PCR (Table 1). Carriage was detected by culture/PCR at day 2, whereas maximum detection by microscopy was evident at day 6. This is not explained by colonisation density and therefore the sensitivity of culture/PCR. This could be explained by greater association with the epithelium (detected by all techniques) rather than the overlying mucus (detected by culture/PCR but not microscopy) over time. Clearance generally occurred between day 9 and day 27 (Table 1 and Fig. 1b).

Curette biopsy samples visualised by confocal microscopy using wheat germ agglutinin (WGA) as a marker for surface carbohydrates and junctional adhesion molecule-A (JAM-A) as a marker of tight junctions, yielded intact sheets of epithelial cells (Supplementary Fig. 1b) associated with polymorphonuclear leucocytes, characterised morphologically (Fig. 1a and Supplementary Fig. 1a). Amongst carriers, pneumococci were found on the epithelial surface (Table 1; Fig. 1c, d, Supplementary Fig. 1b) which over time (typically by 6 days), consisted of two or more pneumococci at the same location on the epithelial surface, which we have termed micro-colonies (Fig. 1e, Supplementary Fig. 1b). There was evidence of what we have termed pneumococcal "micro-invasion" of the epithelial monolayer consisting of either internalisation of bacteria and/or transmigration across the epithelial barrier (Fig. 1e, f). Pneumococci were also observed in phagocytic cells (Supplementary Fig. 1a). Colonisation was frequently characterised by co-association between *S. pneumoniae* and JAM-A (Fig. 1f). This intimate association between inoculated pneumococci and the mucosa during asymptomatic pneumococcal carriage in humans is suggestive of an active engagement process which may influence the outcome of colonisation.

Further visualization of this bacteria–host interaction by transmission electron microscopy has proven problematic in vivo, but we have been able to demonstrate the integrity of the nasal epithelial curette biopsy samples (Supplementary Fig. 1b). We provide evidence of diplococci on the epithelial cell surface and chains or micro-colonies of diplococci that may have been dislodged from the epithelial surface (Supplementary Fig. 1b).

**Epithelial surface marker expression in vivo is unaltered during colonisation of *S. pneumoniae*.** To determine whether pneumococcal colonisation leads to epithelial activation, we analysed nasal curette biopsy cells for surface expression of molecules expressed in airway epithelium which have previously been shown to be upregulated in response to common bacterial stimuli and are involved in immune cell recruitment. These were EpCAM (for epithelial cell identification),

**Table 1 S. pneumoniae association with the nasal epithelium in the EHPC model detected by culture, confocal microscopy and LytA PCR**

| ID | Pneumococcal density | Sample day | | | | | Pneumococcal association | | |
|---|---|---|---|---|---|---|---|---|---|
| | | Pre | D2 | D6 | D9 | D27 | Surface | Intracellular | Lateral |
| 1 | Culture (CFU/ml) | 0 | 373 | 520 | 10 | 2 | | | |
| | Microscopy (counts) | 0 | + | + | 0 | + | + | 0 | 0 |
| | LytA PCR | NEG | POS | POS | n/a | POS | | | |
| 2 | Culture (CFU/ml) | 0 | 0 | 0 | 0 | 0 | | | |
| | Microscopy (counts) | + | + | 0 | 0 | 0 | ++ | + | 0 |
| | LytA PCR | NEG | NEG | NEG | n/a | NEG | | | |
| 3 | Culture (CFU/ml) | 0 | 0 | 0 | 0 | 0 | | | |
| | Microscopy (counts) | 0 | + | 0 | 0 | 0 | + | 0 | 0 |
| | LytA PCR | NEG | POS | NEG | n/a | NEG | | | |
| 4 | Culture (CFU/ml) | 0 | 3 | 9 | 0 | 0 | | | |
| | Microscopy (counts) | 0 | 0 | 0 | + | 0 | + | 0 | 0 |
| | LytA PCR | NEG | POS | POS | n/a | NEG | | | |
| 5 | Culture (CFU/ml) | 0 | $7 \times 10^3$ | 223 | 22 | 0 | | | |
| | Microscopy (counts) | 0 | + | + | + | 0 | + | + | 0 |
| | LytA PCR | n/a | n/a | n/a | n/a | n/a | | | |
| 6 | Culture (CFU/ml) | 0 | $1.9 \times 10^4$ | $5.7 \times 10^3$ | $3.7 \times 10^4$ | 11 | | | |
| | Microscopy (counts) | 0 | ++ | ++++ | ++ | 0 | +++ | ++ | + |
| | LytA PCR | NEG | POS | POS | n/a | POS | | | |
| 7 | Culture (CFU/ml) | 0 | 822 | $1.1 \times 10^3$ | 160 | 0 | | | |
| | Microscopy (counts) | 0 | + | +++ | ++ | 0 | ++ | + | + |
| | LytA PCR | NEG | POS | POS | n/a | NEG | | | |
| 8 | Culture (CFU/ml) | 0 | 0 | 0 | 0 | 0 | | | |
| | Microscopy (counts) | 0 | 0 | 0 | 0 | 0 | 0 | 0 | 0 |
| | LytA PCR | n/a | n/a | n/a | n/a | n/a | | | |
| 9 | Culture (CFU/ml) | 0 | 221 | 695 | 4 | <1 | | | |
| | Microscopy (counts) | n/a | + | +++ | + | n/a | +++ | + | + |
| | LytA PCR | NEG | POS | POS | n/a | NEG | | | |
| 10 | Culture (CFU/ml) | 0 | $1 \times 10^3$ | 0 | 0 | 0 | | | |
| | Microscopy (counts) | n/a | 0 | + | 0 | n/a | + | 0 | 0 |
| | LytA PCR | NEG | POS | POS | n/a | POS | | | |
| 11 | Culture (CFU/ml) | 0 | $1.8 \times 10^6$ | $1.8 \times 10^6$ | $1.9 \times 10^4$ | <1 | | | |
| | Microscopy (counts) | n/a | ++++ | ++++ | + | n/a | +++ | + | + |
| | LytA PCR | NEG | POS | POS | n/a | NEG | | | |
| 12 | Culture (CFU/ml) | 0 | 0 | 0 | 0 | 0 | | | |
| | Microscopy (counts) | n/a | ++ | 0 | 0 | n/a | ++ | + | + |
| | LytA PCR | NEG | NEG | NEG | n/a | NEG | | | |
| 13 | Culture (CFU/ml) | 0 | 23 | 2 | 0 | 0 | | | |
| | Microscopy (counts) | n/a | + | 0 | 0 | n/a | + | 0 | 0 |
| | LytA PCR | POS | POS | POS | n/a | POS | | | |

Nasal washes and nasal curette biopsies were collected from carriage positive and carriage negative volunteers over time. Standard methods for measuring bacterial density by culture (CFU) and LytA PCR were compared against counts visualised by confocal microscopy for pneumococcal association with nasal cells over time. The data were derived from 13 volunteers. +(1–10 pneumococci); ++ (11–50 pneumococci); +++(51–100 pneumococci); ++++ (>100 pneumococci); n/a = reading not taken

IL-22Ra1, HLADR, CD40, CD54 and CD107a (Supplementary Fig. 2a-d). We did not detect a significant change in the relative expression of IL-22Ra1 (protects the epithelial barrier, promotes anti-microbial product secretion during infection, modulating pneumococcal carriage and clearance[24]), HLADR (mediates T cell-antigen recognition and is a marker of epithelial activation[25]), CD40 (co-stimulatory protein which binds CD154[26] (Supplementary Fig. 2e-g)), or CD54 (a leucocyte adhesion molecule which is also upregulated by CD40[26,27], initiating neutrophil migration and recruitment[28], Fig. 2a). However, although numbers of epithelial cells expressing CD107a did not change over time (Fig. 2b, left), we observed an increase in CD107a[high] expression at day 2 post inoculation in carriage positive volunteers vs. carriage negative volunteers (Fig. 2b, right). CD107a in the epithelium forms the membrane glycoprotein of late endosomes and phagolysosomes[16] and has previously been implicated in pneumococcal endocytosis[16]. These data highlight the ability of the pneumococcus to colonise the nasopharynx without marked inflammation and potentially implicate CD107a in the process.

**Adhesion, micro-colony formation and micro-invasion in vitro.** To further understand epithelial responses to S. pneumoniae observed in the EHPC model, we undertook infection experiments with Detroit 562 epithelial cells. We compared the 6B response with two other representative clinical isolates: a 23F strain, TIGR4 (the original sequenced strain[29]) and a TIGR4 pneumolysin deletion mutant strain (dPly).

TIGR4 association with Detroit 562 cells was over 100-fold higher for adhesion and over 70-fold higher for invasion after 3 h infection, compared with 6B or 23F strains (Fig. 3a, b). Using capsule-specific antibody stain for confocal and electron microscopy, pneumococcal adhesion to the epithelial surface at 3 h was found to be associated with micro-colony formation, pneumococcal chain formation and internalisation of this conventionally extracellular pathogen[14] (Fig. 3d–f, j, k, Supplementary Fig. 3a). Internalisation was associated with the formation of epithelial membrane folds (Fig. 3k left). Intracellular pneumococci were contained within intracellular vesicles suggestive of endocytosis, which was most marked with the TIGR4 and dPLY-TIGR4 strains (Fig. 3b, g, k)[16]. Intracellular pneumococci were seen

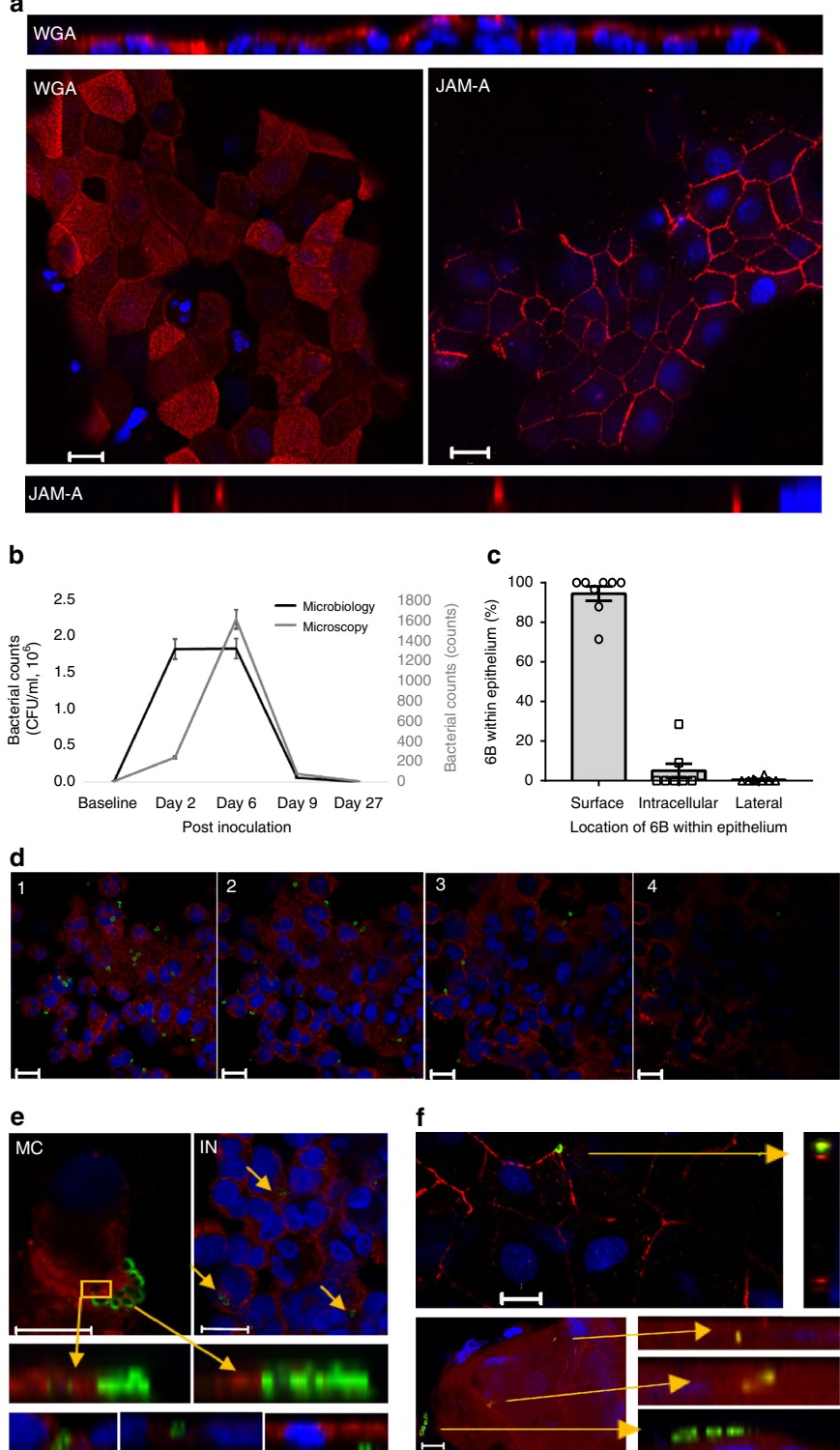

**Fig. 1** Pneumococcal colonisation in the EHPC model is associated with adhesion, micro-colony formation and micro-invasion. **a** Representative fields from the EHPC model nasal curette biopsies show intact epithelium can be obtained from the samples (as visualised by XY planes) and they retain their size and shape (as visualised by XZ planes). Cells were stained with wheat germ agglutinin (WGA) or junctional adhesion molecule-A (JAM-A, red) and nuclei (blue). **b** The pattern of pneumococcal density detected by culture and microscopy. The total bacterial sum across volunteers is shown. **c** The proportion of bacteria located on the cell surface, intracellularly, or paracellularly visualised by confocal microscopy, quantified from eight volunteers collected over time from microscopy counts. **b**, **c** Error bars represent s.e.m. **d** Cells were stained for surface carbohydrates using WGA (red), and the bacteria were marked with specific serotype antiserum (green). XY images of 1 μm slices through a layer of cells from top (stack 1) to bottom (stack 4), with bacteria associated. **e** Examples of cells collected on day 6 post inoculation stained with WGA (red) showing; (MC) micro-colony formation on the surface of the epithelium, (IN) bacterial internalization, (TM) transmigration through epithelial uptake internally or between cells. **f** Co-association between *S. pneumoniae* (green) and JAM-A (red). Nuclei (blue). Bacterial uptake appeared co-associated with JAM-A (yellow). All scale bars represent 10 μm

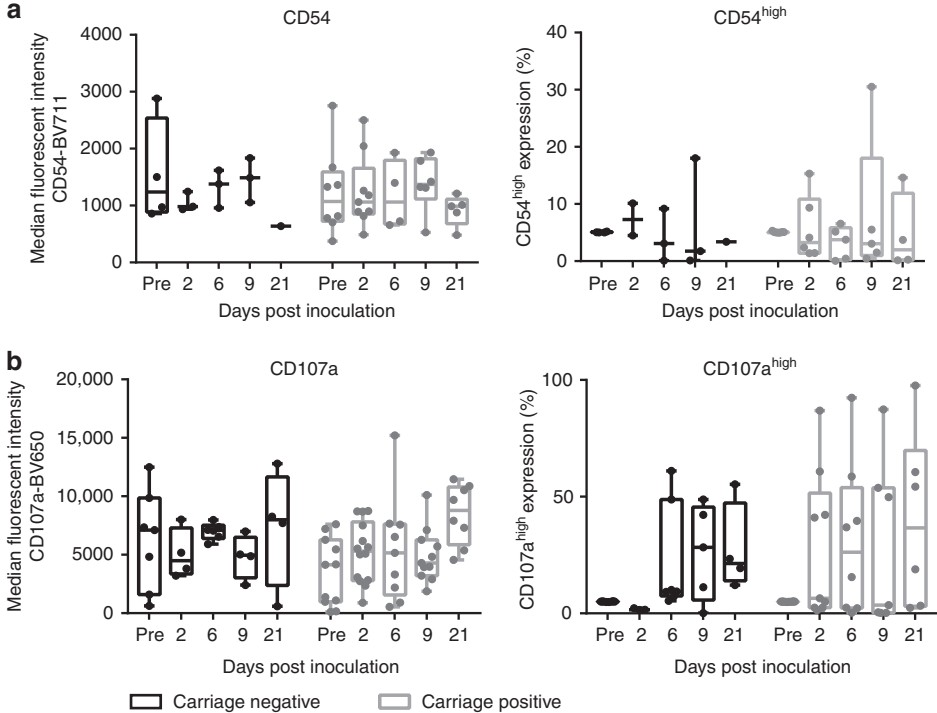

**Fig. 2** Epithelial surface marker expression in vivo is unaltered during colonisation of *S. pneumonia*. Epithelial cells from nasal curette biopsies were analysed by flow cytometry for **a** CD54 and **b** CD107a. Results were expressed as median fluorescence intensity (left) and high cell surface expression (≥95% of the baseline expression, right) representing at least two volunteers per time point who were carriage negative (black) and carriage positive (grey). Comparisons between carriage negative and carriage positive, on each day of sampling, was analysed by one-way ANOVA (**a**—parametric, **b**—non-parametric). $P = 0.84$ CD54 MFI; $P = 0.98$ CD54 %; $P = 0.09$ CD107a MFI; $P = 0.10$ CD107a %. Boxplots represent the median (centre line) and interquartile range (box), minimums and maximums (whiskers)

within vacuoles (Fig. 3k middle), coated with JAM-A (a tight junction protein that regulates barrier function and cell polarity, Fig. 3g, Supplementary Fig. 3a) and associated with the adherens junction protein β catenin (that is involved in epithelial adhesion, Fig. 3g (top right) and Fig. 3h), but not Claudin-4 (a tight junction protein that regulates paracellular permeability, Supplementary Fig. 3a). Pneumococci were located laterally, transiting between cell junctions (Fig. 3h, k right), at the level of the nuclei and, below the basal membrane (Fig. 3i). Transmigration across the epithelium (endocytosis or paracellular movement), as measured by the culture of pneumococci from the basal chamber of transwell inserts, was most marked with 23F (Fig. 3c, fivefold higher compared with TIGR4, after 3 h infection). A similar pattern of epithelial interaction was also seen at 1 h suggesting these observations are not simply explained by differential growth.

Epithelial infection with the dPly-TIGR4 mutant showed a significant increase in adherence (Fig. 3a, tenfold higher after 3 h infection), internalisation (Fig. 3b, threefold higher after 3 h infection, Fig. 3k), but no significant differences in transmigration (Fig. 3c) compared with the wild-type strain, implicating pneumolysin in the modulation of bacterial interactions with the epithelium.

Similar patterns of adherence and invasion were observed in infection experiments with human Calu3 and A549 epithelial cells demonstrating that these findings are cell-line independent (Supplementary Fig. 4). Furthermore, primary epithelial cells differentiated on an air-liquid interface for 30 days and co-cultured with either 6B or 23F*S. pneumoniae* also revealed pneumococcal micro-colony formation, micro-invasion and epithelial junctional protein association (Supplementary Fig. 1c).

Together these data show the strain dependency of pneumococcal–epithelial interactions in vitro which mirror the epithelial micro-invasion (both internalisation and/or transmigration of pneumococci) seen in vivo.

**Endocytosis-related micro-invasion and transmigration**. To confirm that the pneumococcal endocytosis leads to transmigration across the epithelium, endocytosis was inhibited with dynasore and nystatin[16]. After 1 h of infection, cellular uptake of 23F was inhibited by 82% (Fig. 4a) and transmigration across the cell monolayer by 85% (Fig. 4b). In agreement with previous studies using A549 and Calu3 cells[30], we found that viable pneumococci in Detroit 562 cells decreased over time (Supplementary Fig. 5a). Using gentamicin treatment to eliminate the original extracellular inoculum, we observed *S. pneumoniae* in the apical and basal media (Supplementary Fig. 5b), suggesting egress from the epithelium.

**Micro-invasion does not compromise epithelial barrier function**. *S. pneumoniae* has been shown to affect the integrity of epithelial barriers and tight junction function in murine models[13,31]. Having demonstrated pneumococcal micro-invasion without marked epithelial morphological disruption both by confocal and electron microscopy, we explored possible effects on epithelial barrier function by measuring trans-epithelial electrical resistance (TEER) and permeability to 4 kDa FITC-dextran in Detroit 562 cells. TEER was not adversely affected by pneumococcal co-culture (Fig. 4c) and permeability was maintained with all the pneumococcal strains at 3 h post-infection (Fig. 4d). Although co-localisation of pneumococci was seen with host proteins, we did not detect marked epithelial membrane

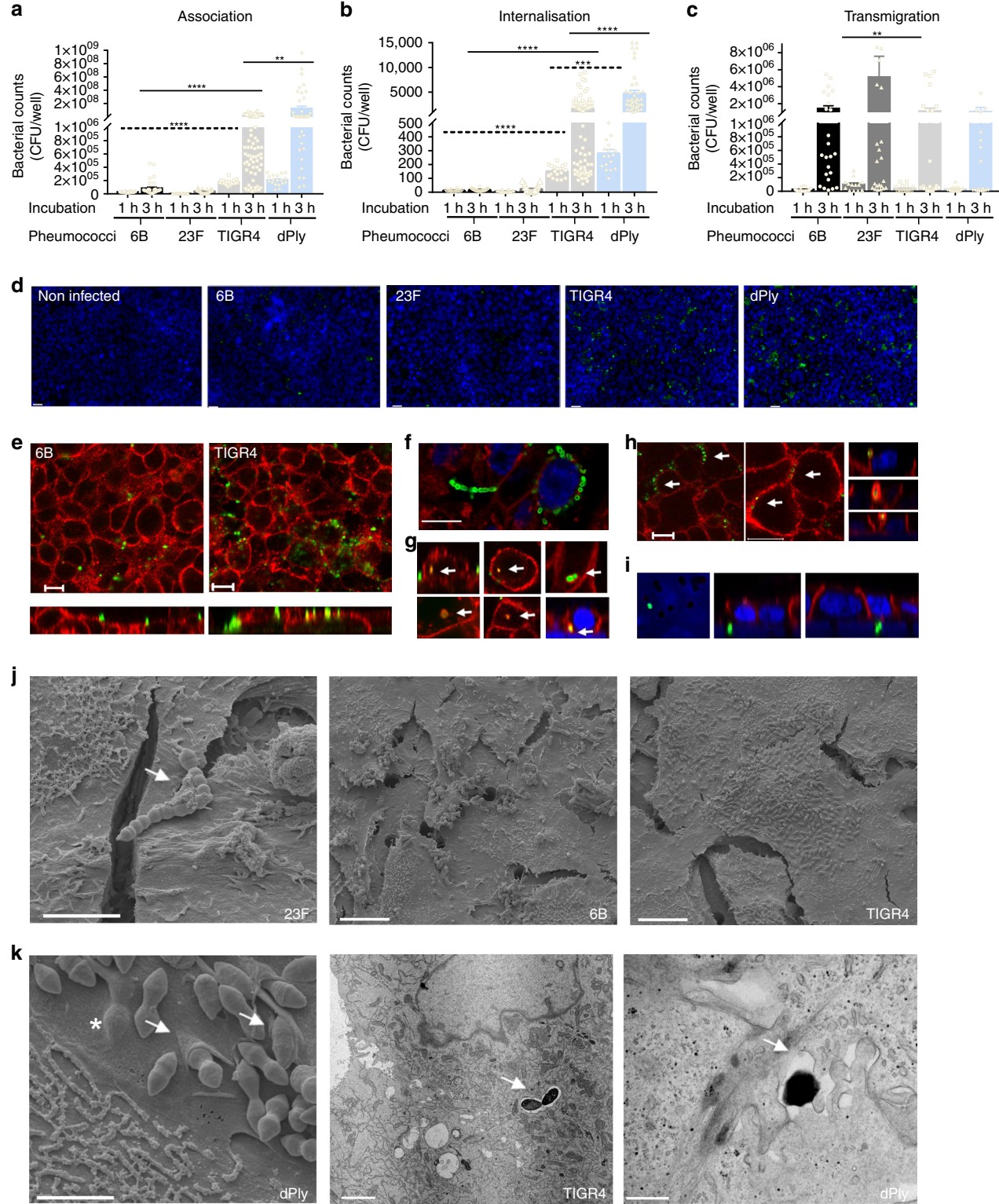

carbohydrate or protein re-distribution (WGA, Claudin-4 JAM-A, β catenin, Supplementary Fig. 3a). Similar observations were made with A549 cells and Calu3 cells (Supplementary Fig. 4). These data suggest that pneumococcal micro-invasion was not dependent on loss of epithelial cell barrier function during early interactions.

**S. pneumoniae upregulation of epithelial CD54 and CD107a in vitro.** In a similar manner to primary cells collected from nasal curettes, we assessed Detroit 562 cell surface marker expression in response to S. pneumoniae (Supplementary Fig. 6). There were no significant changes in IL-22Ra1, HLADR or CD40 expression in response to 6B, 23F, TIGR4 and dPly-TIGR4 strains (Supplementary Fig. 6d–f). However, CD54high expression was significantly greater in response to the TIGR4 and dPly-TIGR4 strains, compared with non-infected cells (Fig. 5a). Epithelial CD107a was upregulated in response to the 6B, 23F and TIGR4 strains (Fig. 5b), but this was not seen with dPly-TIGR4.

**Fig. 3** Pneumococcal infection in vitro is associated with adhesion, micro-colony formation and micro-invasion. Pneumococcal **a** association and **b** internalisation: Detroit 562 cell monolayers were stimulated with pneumococci for 1 or 3 h and the quantity of associated bacteria was determined by culture (CFU); $n = 6$. ****$P = < 0.0001$, ***$P = 0.0001$. **a** 1 h (dashed lines, ANOVA); 3 h (solid lines, Kruskall–Wallis); **b** (Kruskall–Wallis). **c** Pneumococcal transmigration: Cells on transwell inserts were stimulated with pneumococci for 3 h and bacterial density in the basal chamber was determined (**$P = 0.0092$, Kruskall–Wallis). $N = 5$. **a–c** Error bars represent s.e.m. **d** Representative pneumococcal-density (green) images of cells (blue nuclei). Scale bar = 20 μm. $n = 5$. **e–h** Representative images of pneumococcal localization (green) from cells (red, JAM-A), illustrating: **e** differences in adherence by 6B and TIGR4; **f** micro-colony and chain formation (green) dPly-TIGR4 (green) and WGA (red); **g** internalised bacteria (top 6B; bottom TIGR4) co-localised with JAM-A with associated intracellular vesicle-like bodies (yellow), or co-associated with β catenin, top right; **h** lateral localisation of pneumococci (XY images, TIGR4 (green) and JAM-A (red)) with possible paracellular movement of 6B (green) co-associated with β catenin (red) (XZ images); **i** basal localisation of bacteria (23F, green) at the level of nuclei (blue) and insert pores $N = 5$ with replicates. Scale bar = 10 μm. **j** Scanning EM images of Detroit 562 cells infected with *S. pneumoniae* which appear as diplococci. Left—pneumococcal chain formation (strain 23F, arrow, scale bar = 4 μm); middle and right—micro-colony formation by strains 6B and TIGR4 on the epithelial surface (scale bar = 10 μm). **k** Micro-invasion of dPLY-TIGR4 shown by EM: left—scanning EM showing epithelial membrane folding (arrows) and pneumococci below the cell membrane surface (* scale bar = 2 μm); middle—internalisation of TIGR4 pneumococci encased within a vacuole (arrow, scale bar = 2 μm); and right—transmission EM showing transmigration of dPLY-TIGR4 pneumococci between epithelial cells (arrow, scale bar = 0.5 μm)

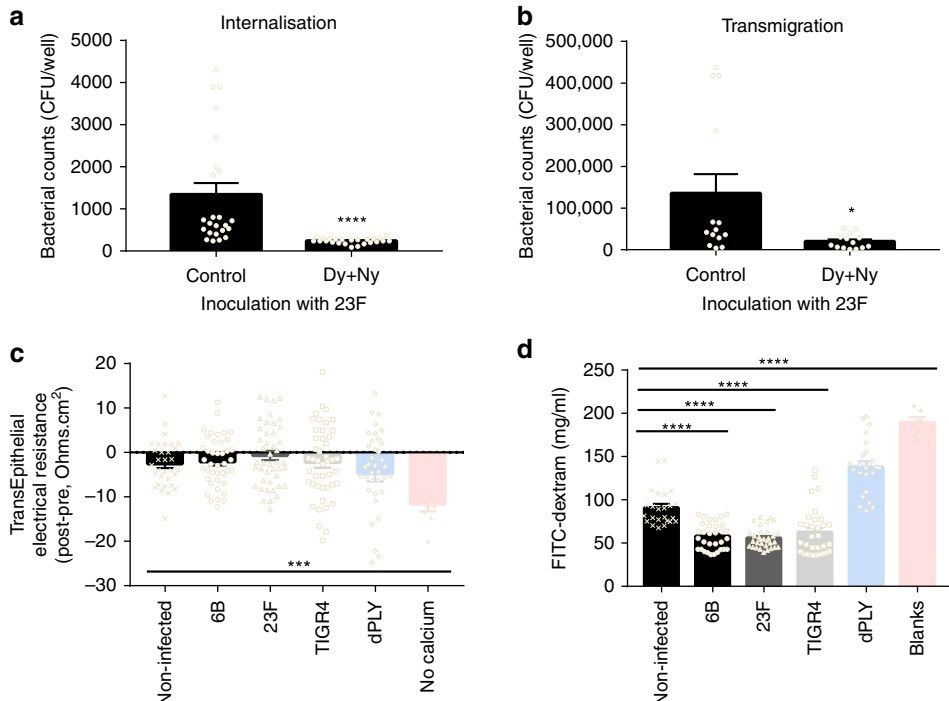

**Fig. 4** Pneumococcal micro-invasion is endocytosis-dependent and occurs without loss of barrier function. **a**, **b** Detroit 562 cells were treated with dynasore and nystatin for 30 min prior to and during incubation with 23F, for 1 h. **a** Intracellular bacteria were quantified using a gentamicin protection assay and CFUs counted (****$P = < 0.0001$, Mann–Whitney). **b** Transmigrated bacteria after 1 h were counted by CFUs (*$P = 0.013$ Mann–Whitney). $N = 4$. **c**, **d** Cells were co-infected with pneumococci for 3 h on transwell inserts and barrier function was measured. **c** Trans-epithelial electrical resistance (TEER) was compared before and after exposure to pneumococci. $N = 9$. Calcium withdrawal was used as a positive control (***$P = 0.0013$, unpaired T-test, $n = 3$). **d** Permeability was assessed by leak to 4 kDa FITC-dextran, collected from basal chamber. $N = 4$. ****$P = < 0.0001$ comparing non-infected against pneumococci (Kruskall–Wallis). Blank inserts were used as a positive control (Mann–Whitney). Error bars represent s.e.m.

These data implicate pneumolysin in the induction of CD107a but not CD54 epithelial surface expression and support the possibility that CD107a is involved in pneumococcal–epithelial micro-invasion events seen in vivo.

**S. pneumoniae induces epithelial secretion of IL-6, IL-8 and CD54 in vitro.** To assess the inflammatory consequences of pneumococcal–epithelial infections in vitro, we measured a cytokine panel in Detroit 562 cell supernatants following *S. pneumoniae* incubation (Fig. 5c). We detected a significant increase in IL-6 and IL-8, which were strain dependent and partially pneumolysin dependent. In line with surface marker observations, only TIGR4 significantly upregulated the secretion of soluble CD54, which was dependent on pneumolysin (Fig. 5c).

These data highlight the less inflammatory nature of the 6B strain compared with the more invasive strain, TIGR4, and the importance of pneumolysin in the epithelial response. Pneumococcal wild-type strain differences were not explained by differences in pneumolysin activity, as determined by haemolysis (Supplementary Fig. 7).

**Epithelial innate transcriptomic responses to infection in vitro.** To explore the hypothesis that the pattern of epithelial adhesion and micro-invasion results in differential epithelial responses, we performed RNAseq of Detroit 562 cells infected with our panel of *S. pneumoniae* strains. In comparison with non-infected epithelial cell cultures, infection with TIGR4 upregulated expression of 1127 genes (517 unique genes), 23F upregulated 650 genes (69

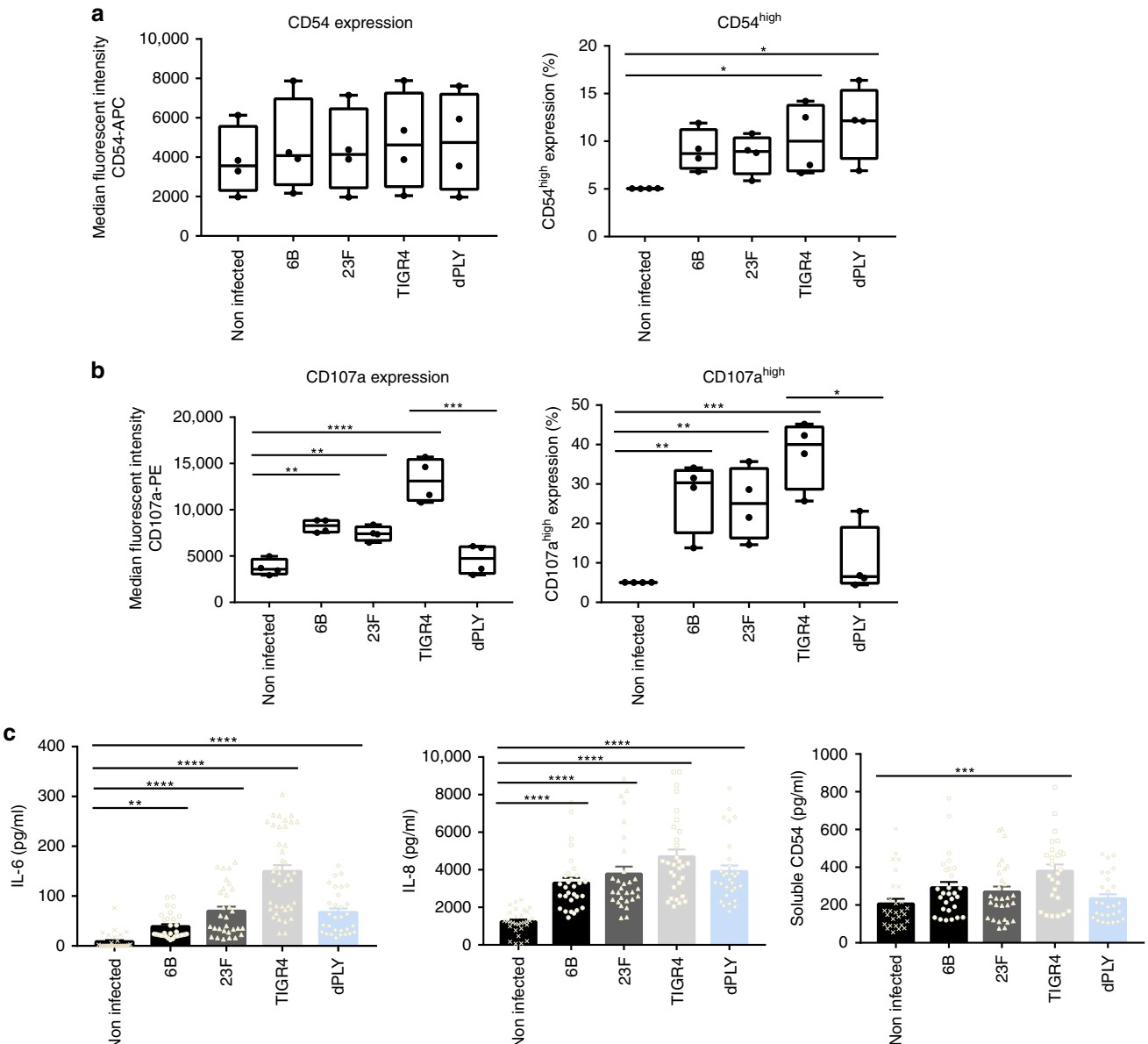

**Fig. 5** Epithelial innate activation following pneumococcal infection in vitro. **a**, **b** Detroit 562 cells were stimulated with *S. pneumoniae* for 6 h and the median fluorescence intensity and high-expressing cells for (**a**) CD54 and (**b**) CD107a were analysed by flow cytometry. $n = 3$ independent experiments (****$P <$ 0.0001 CD107a MFI, compared with non-infected cells, ANOVA. ***$P = 0.0009$ for MFI CD107a, TIGR4 vs dPly, unpaired T-test. ***$P < 0.001$ CD107a high expression, compared with non-infected cells, ANOVA. *$P = 0.028$ for high expression CD107a, TIGR4 vs dPly, Mann–Whitney). (*$P = 0.048$ NI vs TIGR4; *$P = 0.0367$ NI vs dPly for CD54, ANOVA). Boxplots represent the median (centre line) and interquartile range (box), minima and maxima (whiskers). **c** IL-6, IL-8 and CD54 in the supernatants from Detroit 562 cells stimulated with *S. pneumoniae* for 6 h, were measured by ELISA. $N = 6$ independent experiments with replicates (****$P < 0.0001$ IL-6; ***$P = 0.0013$ CD54; ****$P < 0.0001$ IL-8 (Kruskall–Wallis test)). Error bars represent s.e.m.

unique genes), and 6B upregulated 153 genes (10 unique genes) (Fig. 6a). The pneumolysin deficient isogenic mutant of TIGR4 upregulated 220 genes (14 unique genes). Compared with non-infected cells, 1193 genes were upregulated by *S. pneumoniae* infection overall and 93 were core to all strains tested (differentially upregulated genes in Supplementary Data 1 and Reactome pathways in Supplementary Data 2).

To elucidate the pathways that drive the transcriptional responses to each strain, we also undertook transcription factor binding site enrichment analysis (Fig. 6b, transcription factor binding site enrichment analysis in Supplementary Data 3). The responses to 6B and pneumolysin deficient strains were principally enriched in binding sites for the NFkB/Rel family of transcription factors, indicating that an NFkB activation pathway

was the dominant driver for responses to these strains. In contrast, cellular responses to TIGR4 and 23F strains revealed enriched binding sites for more diverse transcription factors, suggesting broader molecular pathways by which these strains may influence cellular function. This analysis revealed particular enrichment of binding sites for beta-beta-alpha zinc finger superfamily of transcription factors, including KLF4, suggesting that these strains upregulate mitogen activated protein kinase pathways upstream of these transcription factors[32].

The sets of genes upregulated by each strain individually were subjected to core and interactome and pathway-enrichment analysis for functional annotation of the cellular response at systems level (Fig. 6c). From 20 clusters of pathways, significant enrichment of innate immune system responses was evident in all

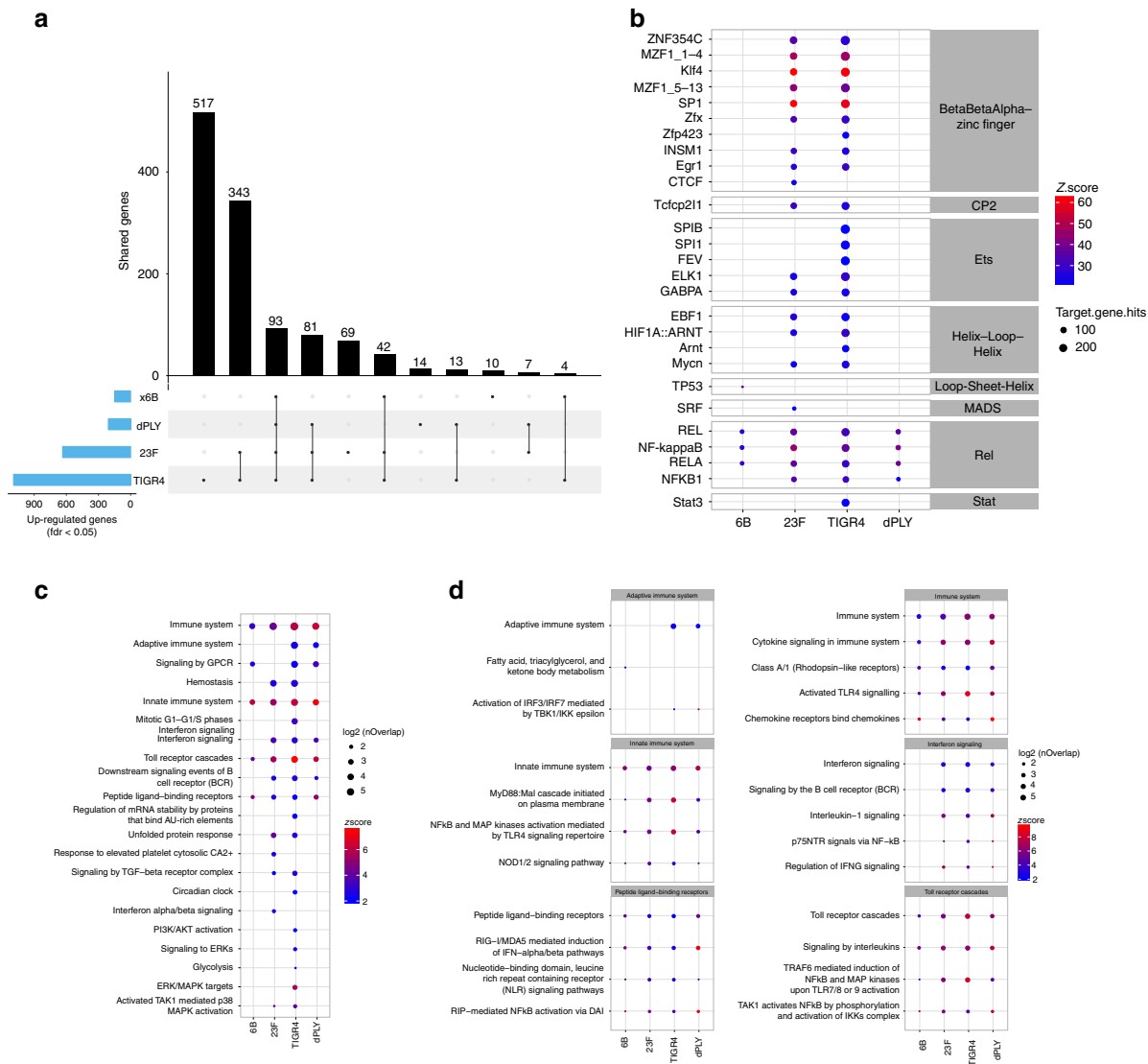

**Fig. 6** Transcriptomic responses to *S. pneumoniae* in vitro are influenced by adhesion and micro-invasion. **a** The total number of Detroit 562 epithelial genes upregulated following exposure to 6B, 23F, TIGR4 and TIGR4-dPly pneumococci, compared with non-infected samples. $N = 3$ independent experiments with replicates. Genes with an FDR <0.05 were considered for further analysis. The matrix shows intersections for the four strains, sorted by size. Dark circles in the matrix indicate sets that are part of the intersection. **b** Bubble heatmap showing the results of transcriptional factor analysis in Opossum for each strain. Transcriptional factors are ordered by family and then sorted by number of genes in the database. The colour represents the log2 z-scores (a continuum from blue (lowest values) to red (highest values)) and the size shows the number of genes annotated to the particular pathway. **c** Bubble heatmap showing the results of reactome pathway analysis for upregulated genes in each strain compared with non-infected cells. Pathways were clustered in 20 groups using K-means on Jaccard index and the biggest pathway for each group was selected for representative annotation of each cluster. Individual pathways within clusters are shown in Supplementary Fig. 8. The colour represents the log2 z-scores (a continuum from blue (lowest values) and red (highest values)) and the size shows the number of genes annotated to each pathway. **d** Selected bubble heatmap clusters representing the individual pathways within each cluster from (**c**)

strains and included specific enrichment in Toll receptor cascades, peptide ligand-binding receptors, NFkB and MAP kinase activation, Interferon signalling and cytokine and chemokine signalling (Fig. 6c, d, Supplementary Fig. 8). In general, each pneumococcal strain induced a distinct transcriptomic profile, with TIGR4 and 23F strains inducing a broader number of biological pathways than the 6B strain (Fig. 6c). Fewer transcriptional responses at the level of functional pathways in response to dPly-TIGR4 is consistent with a role for pneumolysin in innate immune recognition of *S. pneumoniae*[33]. Fewer transcriptional responses to 6B is consistent with our observations that this strain exhibits the lowest level of cellular association and transmigration.

**Epithelial transcriptomic responses to 6B are detectable in vivo**. Next, we sought to test whether these epithelial transcriptomic responses to *S. pneumoniae* were also evident in the 6B EHPC model. We compared nasal transcriptomes sampled at 2 or 9 days after inoculation with pre-inoculation samples. These revealed enrichment of 162 transcripts among individuals who became colonised (differential gene expression in Supplementary Data 4). These genes included Claudin 5 and Claudin 17, Defensin β 103A/B, Cadherin 16, Desmocollin 1 and Gap junction protein α1, which suggests cytoskeletal re-organisation 2 days post inoculation in carriage positive individuals. By day 9 post inoculation, molecules such as CCR3, matrix metalloproteinase 12 and MHC II molecules were enriched, suggesting

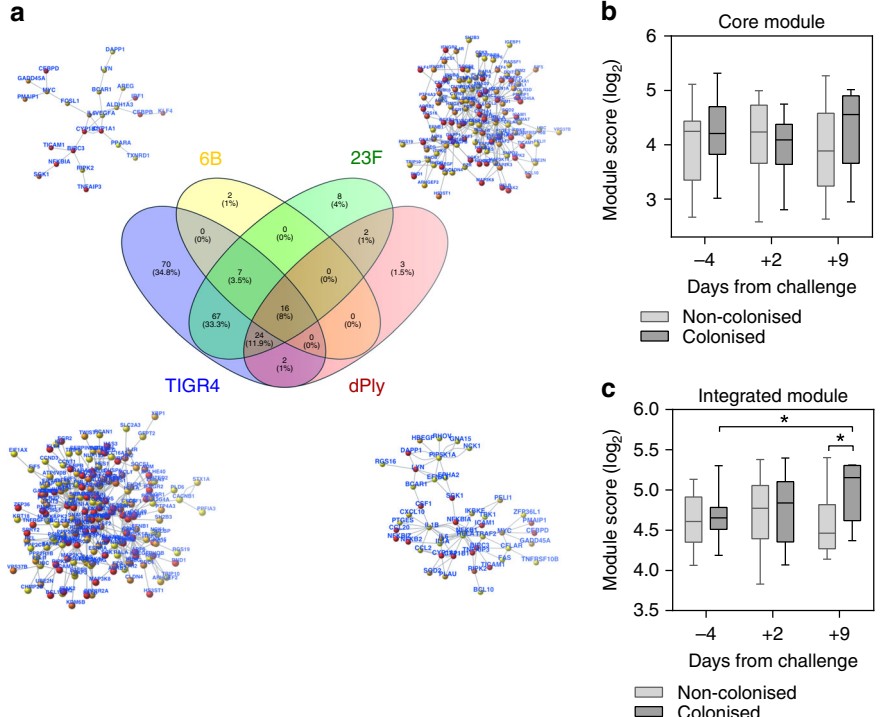

**Fig. 7** Epithelial transcriptomic responses coincides with bacterial clearance in vivo. **a** Network representation generated from the in vitro analyses for each strain. From the genes present in the networks, an integrated gene list (200 genes) and a core genes list (16 genes) were generated and tested as modules in the in vivo data. **b**, **c** Upregulated genes in comparison to the transcriptomes of carriage negative and carriage positive volunteers were compared between time points. Plots represent log2 TPM arithmetic mean for genes in the core (**b**) and integrated (**c**) modules. $P = < 0.05$, (one tail Mann–Whitney test). Number of samples for carriage negative and positive, respectively: pre (17:14), day 2 (15:14), day 9 (7:7). Boxplots represent the median (centre line) and interquartile range (box), minima and maxima (whiskers)

activation of the nasal mucosa[34]. However, these genes were not significantly enriched for any immune response pathway annotation perhaps because of the statistical stringency necessary for genome-wide multiple testing and were not epithelial-specific.

Gene expression modules can successfully detect specific transcriptional programs in bulk tissue[35]. Therefore, as an alternative approach, we sought to target specific epithelial responses in vivo using the in vitro interactome. A direct comparison of transcripts for 6B revealed FOSL-1, an AP-1 transcription factor subunit, to be enriched with carriage, which has previously been reported to become activated following pneumococcal challenge in BEAS-2B and HEK293 cells[36]. To expand the sensitivity of the approach, we evaluated the expression of two transcriptional modules in the nasal samples (Fig. 7). A 'core' module comprised of 16 interacting genes (Fig. 7b) that were upregulated in the Detroit epithelium by all the pneumococcal strains from Fig. 7a, and a second 'integrated' module that comprised of 200 interacting genes (Fig. 7c), which were upregulated in the Detroit epithelium by at least one of the pneumococcal strains from Fig. 7a. Among carriage negative individuals, we found no increase in expression in either gene expression modules (Fig. 7b, c). In contrast among the individuals who became colonised, the increased expression of both modules (significantly increased within the integrated module, Fig. 7b, c), suggests that the prominence of responses at day 9 may reflect time-dependent proliferation of the 6B bacterial inoculum, epithelial engagement and the subsequent epithelial activation. These responses coincide around the time of bacterial clearance.

## Discussion

It is widely accepted that colonisation of the URT mucosa by many potentially pathogenic bacterial species, involves transient

association with the overlying mucus layer but that adherence to the epithelial surface avoids mucus entrapment and muco-ciliary clearance[14,37]. While evading immunity, bacterial replication then occurs prior to onward transmission to a new host. Even for S. pneumoniae, the mechanisms underlying this transition from acquisition to established colonisation, and then transmission and/or disease, are not well understood. By combining an EHPC model and in vitro human cell culture systems, we show that pneumococcal colonisation leads to epithelial adherence, micro-colony formation and migration across the epithelial barrier without disease, which we have termed micro-invasion. The finding of co-association with junctional proteins provides a possible mechanism for micro-invasion. Clearance of the pneumococcus from the mucosa in vivo occurs in the context of micro-invasion. Indeed, epithelial micro-invasion and not bacterial load per se, appears to dictate unique transcription factor binding site signatures, downstream cytokine, chemokine and metabolic pathway enrichment (Fig. 8).

The concept of mucosal micro-invasion moves us away from current models of colonisation where the pneumococcus is held up at the epithelial surface and that invasion leads to disease. Our data are supported by murine colonisation experiments[38,39] and the detection of pneumococcal DNA in blood of otherwise healthy, colonised children[40]. In line with other cell culture and murine models[13,16,38,41], we have demonstrated that pneumococcal micro-invasion occurs by endocytosis through the formation of cytoplasmic vacuoles, and by paracellular transcytosis across the epithelium. This process appears to be association with the formation of epithelial folds different from the membrane ruffles previously reported with conventionally intracellular pathogens such as Salmonella typhimurium[42]. Our study emphasises the active nature of these processes without marked

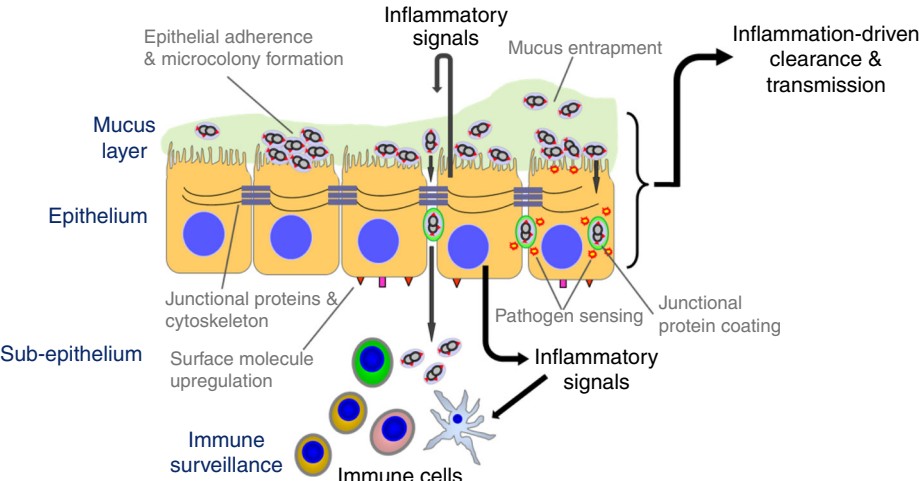

**Fig. 8** Model of pneumococcal colonisation at the human mucosal epithelium. Pneumococcal adhesion and micro-colony formation on the epithelial surface may lead to micro-invasion; internalisation of the bacteria and/or transmigration across the epithelial barrier (micro-invasion). The epithelial-derived response is dependent on the subsequent pattern of interactions. Micro-invasion amplifies epithelial sensing and inflammation/innate immunity, which we postulate leads to immune cell engagement. This process of epithelial sensing inflammation/innate immunity may enhance both clearance and transmission. Co-association with junctional proteins may facilitate migration across the barrier

cellular damage or loss of barrier function. Micro-invasion of the epithelium may overcome the inaccessibility of PAMP receptors located either at the epithelial basolateral surface or intracellularly[43,44]. We have found that at early time points, micro-invasion in vitro is also characterised by pneumococcal co-association with JAM-A and β catenin without evidence of barrier dysfunction. In murine models at later time points, TLR-dependent Claudin 7 and Claudin 10 downregulation enhanced pneumococcal translocation across the epithelium has been observed[13]. While this may occur as inflammation becomes more prominent, our data suggest that micro-invasion can happen without barrier disruption and that the epithelium plays an active role in the regulation of these complex host–pathogen interactions[12]. How micro-invasion relates to the risk of invasive pneumococcal disease remains to be determined but as demonstrated by the EHPC model, micro-invasion may occur in healthy individuals without symptoms.

The differential impact of micro-invasion on the epithelial transcriptomic response was striking with considerable enrichment of multiple diverse transcription factors and signalling pathways related to innate immunity with the most invasive pneumococcal strains. The interactome module highlights that although enrichment of gene activation in response to all pneumococcal strains are apparent (the core module), a more diverse range of epithelial gene activation occurs between strains, reflecting specific host–pathogen interactions. Nasal colonisation in murine models is proinflammatory[45,46], but our in vitro and EHPC experiments with serotype 6B suggests this is not always the case, whereby pneumococcal-host cell adherence, growth, endocytosis and paracellular migration can be established without a marked host response. In the EHPC, clearance occurred around the time the epithelial transcriptomic response was most prominent. Weiser and colleagues argue that the colonising pneumococcus induces a host inflammatory response that mediates clearance, but also promotes nutrient acquisition, mucus production and onward transmission[14,18,47], which is supported by murine models[48] and epidemiological studies of viral-co-infection[49]. It has been suggested that pneumolysin induces neutrophil influx and degranulation, leading to increased secretions from the nasopharynx, so promoting

transmission[5,18]. We therefore suggest that the epithelial innate response resulting from pneumococcal colonisation could promote both clearance and transmission.

The enrichment of binding sites for the transcription factor KLF4 transcription factors seen in our in vitro data suggests a counter-regulatory process aimed at minimising inflammation. KLF4 plays a role in barrier function, suppression of NFκB-dependent IL-8 secretion, regulation of IL-10 expression via TLR9-MyD88 and Yes-1, in response to S. pneumoniae, which is partially dependent on autolysin LytA[32,50,51]. Thus, by exploiting micro-invasion, the pneumococcus may carefully calibrate the host innate immune/inflammatory response to promote survival through transmission.

A range of pneumococcal PAMPs including pneumolysin, may trigger this epithelial sensing process[18,52]. We found pneumolysin to be a prominent inducer of epithelial surface molecule upregulation, cytokine production and transcriptomic inflammatory response in vitro. Following internalisation of the pneumococcus by neutrophils, pneumolysin has been shown to induce ROS following bacterial autolysis, which leads to cellular activation[53]. We speculate that pneumolysin released intracellularly may signal directly in epithelial cells or through host cell pore-formation leading to entry of other PAMPs. Mediated by autolysin, the pneumococcus undergoes autolysis during stationery growth phase, resulting in additional PAMPs' release including bacterial DNA[53]. In mice, pneumococcal DNA triggers inflammation through a DAI/STING/TBK1/IRF3 cascade[45,54], a response that may contribute to pneumococcal clearance. In our in vitro transcriptomic analyses, we detected enriched activation of IRF3/IFR7 in dPly-TIGR4. We suggest that in the context of micro-invasion, pneumococcal DNA may also act as a trigger of epithelial sensing, inducing inflammation[54].

Our findings are limited by the pneumococcal strains available for testing in the EHPC model to enable direct comparisons with in vitro data. Nonetheless, the use of different strains in vitro has enabled us and others[55,56] to determine the impact of different patterns of epithelial adherence and invasion on the host inflammatory/innate immune response. Currently, the EHPC model does not allow us to definitively assess human to human transmission although the appropriate methods are under development.

In conclusion, our data implicate the novel finding of micro-invasion during pneumococcal colonisation of otherwise healthy humans, promoting epithelial-derived innate immunity/inflammation and ultimately clearance. The pathways critical for onward pneumococcal transmission remain to be determined in humans but based on murine models, we propose that this epithelial response may also facilitate transmission. The balance between innate immunity/inflammation-driven transmission and clearance may be further modulated by the frequency of carriage events, pneumococcal strain co-colonisation, viral co-infections and other environmental pressures[1,22,48]. Our approach combining in vitro with in vivo human systems offers a potentially tractable way to further interrogate these processes.

## Methods

**Bacteria**. *S. pneumoniae* clinical strains used were 6B (BHN 418[57]) and 23F (P1121[58]) and TIGR4 (P1672[59]), together with a pneumolysin deficient TIGR4 mutant strain (P1672, a kind gift from Prof T Mitchell, University of Birmingham). Stocks of bacterial aliquots grown to O.D.$_{600nm}$ 0.3 were stored at −80 °C, defrosted, resuspended in cell culture media and used once. Colony forming units (CFU) were counted on horse blood agar plates (EO Labs).

**EHPC model**. Following written informed consent, healthy non-smoking adults between the ages of 18 and 59 were inoculated with 80,000 CFU/nostril live 6B *S. pneumoniae* (BHN418), grown to mid-log phase in vegetone broth[60]. Individuals naturally colonised with *S. pneumoniae* or in regular contact with at-risk individuals were excluded. Blood and mucosal T cell and antibody immunity were not systematically measured in this cohort but previous studies of volunteers from this population have demonstrated background T cell and antibody-mediated anti-pneumococcal immunity, but a relationship between pre-challenge immunity and the development of carriage in the model has not been established[21,23]. To exclude the influence of viral-co-infection, EHPC model volunteers who had coryzal or flu-like symptoms prior to challenge were excluded. Volunteers were screened for the presence of 20 common respiratory viruses before inoculation. In total, 1/13 asymptomatic volunteers (volunteer 1) was positive for enterovirus. Nasal washes and mucosal cells (curette biopsy) from the inferior turbinate were obtained by PBS syringe and curettage using a plastic Rhino-probe™ (Arlington Scientific), in this order of collection, repeatedly, before pneumococcal inoculation. Sampling was then repeated on days 2, 6, 9, and 14–27 post pneumococcal inoculation[61]. Volunteers received the TIV vaccine intramuscularly at day 3 post pneumococcal inoculation. Bacteria collected from nasal washes were quantified by CFU counts. At each time point, two curettage samples were obtained and processed for confocal immunofluorescence, flow cytometry, primary cell culture and/or transcriptomic analysis by RNA sequencing (RNAseq). There were no adverse events. The carriage status of each volunteer was blinded until sample analysis was completed.

Ethical approval was given by NHS Research and Ethics Committee (REC)/ Liverpool School of Tropical Medicine (LSTM) REC, reference numbers: 15/NW/ 0146 and 14/NW/1460 and Human Tissue Authority licensing number 12548.

**Ex vivo cell culture**. Primary epithelial cells were cultured according to Müller et al.[62] with modification. Briefly, nasal cell curette biopsy samples were collected in RPMI containing Penicillin-Streptomycin (1%) and FBS (10%) before centrifugation at 400 × *g* for 5 min. Cells were resuspended in BEGM-AS (25 ml PneumaCult BEGM (Bronchial Epithelial Growth Media, Stem Cell Technologies), 25 ml BEGM (Cell Application), GA-1000 (1:2000, Lonza)) supplemented with 50 ng/ml DNase I, incubated for 20 min at RT and centrifuged. The resuspended pellet containing BEGM-S++ media (50 ml PneumaCult BEGM media, sodium bicarbonate (1:50, Fisher Scientific) Nu-serum IV (1:20), GA-1000 (1:1000)), was plated into 12-well cell culture plates (Corning) that had been coated with 300 μg/ml PureCol (Cell System) for 30 min at 37 °C and washed with Hanks Buffer Saline Solution (HBSS$^{+/+}$, with calcium and magnesium, Gibco). 24 h after sampling, media was replaced with BEGM-S++ and BEGM-AS+ (50 ml of BEGM-AS media, 2 nM retinoic acid, GA-1000 (1:2000)) and applied as follows: 24 h—1:1; 48 h—1:3, 72 h and following—1:6. At day 7, the cells were dislodged using 0.25% trypsin, placed in a prepared 15 ml flacon containing Soy trypsin inhibitor (Sigma Aldrich), centrifuged (4 °C, 400 × *g* for 5 min) and resuspended in BEGM-AS+ and seeded into a cell culture flask (Corning). Cells were cultured for a maximum of 20 days. At day 28 (week 4) cells were seeded on Corning 12 well inserts and cultured in BEGM ALI media (25 ml DMEM high glucose, 25 ml BEGM-AS media, 50 μg/ml Bovine Pituitary Extract, 66 μg/ml Nystatin (Fisher Scientific), 1.5 μg/ml BSA, GA-1000 (1:4000)). Cells were seeded at a concentration of $1 × 10^5$ cells/well and cultured until confluent. Apical and basolateral media was replaced with BEGM ALI media supplemented with 500 nM retinoic acid for 48 h. Media was then replaced with PneumaCult-ALI media on the basal side only. Cells were

maintained for 30 days with the apical cell surface rinsed weekly with 10 min of HBSS$^{+/+}$ and removing liquid thereafter.

**Human respiratory tract epithelial cell lines**. Human pharyngeal carcinoma Detroit 562 epithelial cells (ATCC_CCL-138) and human bronchial carcinoma Calu3 epithelial cells (ATCC_HTB-55) were grown in 10% FCS in alpha MEM media (Gibco). Human alveolar epithelial carcinoma A549 epithelial cells (ATCC_CCL-185) were grown in 10% FCS with 1% L-glutamine in Hams/F-12 media (Gibco).

**Pneumococcal–epithelial cell co-culture**. For association and invasion assays, confluent Detroit 562 (typically day 8 post plating), Calu3 (typically day 10 post plating) and A549 (typically day 4 post plating) monolayers cultured on 12 well plates (Corning), were exposed to *S. pneumoniae* for 3 h in 1% FCS alpha MEM (MOI ~1 cell:10 pneumococci). The medium was removed, and cells washed three times in HBSS$^{+/+}$. Cells were incubated in 1% saponin for 10 min at 37 °C and lysed by repetitive pipetting. Dilutions of bacteria were plated on blood agar and colonies counted after 16 h. To quantify internalised bacteria, 100 μg/ml gentamicin was added for 1 h to the cells, which were washed another three times, before incubating with 1% Saponin and plating on blood agar plates. CFUs were counted after 16 h incubation at 37 °C, 5% $CO_2$. There were no differences in pneumococcal pre- or post-inoculum, or growth density between the strains, in the cell supernatant 3 h post-infection.

For transmigration assays, Detroit 562 cells were cultured on 3 μm pore, PET Transwell Inserts (ThermoFisher) typically for 10 days to achieve confluent, polarised monolayers. Calu3 cells were plated onto Transwell inserts typically for 12 days and A549 cells typically for 6 days. Cell culture media was changed 1 h prior to addition of bacteria to 1% FCS (250 μl apical chamber, 1 ml basal chamber). Resistance was recorded before and after *S. pneumoniae* were added using an EVOM2 (World Precision Instruments). In total, 1 mg/ml FITC-dextran (Sigma Aldrich) was added to the apical chamber of selected inserts to assess permeability. Approximately 12 million bacteria were added to the cells (~MOI 1 cell: 25 bacteria). During the time course, 50 μl was removed, diluted and plated, from the basal chamber to measure bacterial load by counting CFU/well. Permeability was recorded using a FLUOstar Omega (BMG Labtech) at 488 nm.

For endocytosis inhibition assays, Detroit 562 cells cultured on 12-well plates were treated with 100 μM dynasore (Cambridge Biosciences) and 7.5 μg/ml nystatin (Sigma Aldrich) to block endocytosis for 30 min prior to, and for the duration of pneumococcal infection incubation period. DMSO was used as a control. Cells were washed and treated with gentamicin and lysed with saponin as described above.

**Confocal microscopy**. For the in vivo analysis, mucosal cells derived by curettage from the EHPC model were placed directly into 4% PFA for 1 h. Cells were cytospun onto microscope slides and allowed to air dry. For the in vitro analysis, epithelial cell lines on transwell membranes were fixed in either 4% PFA (Pierce, Methanol Free) or 1:1 mix of methanol:acetone for 20 min. For both in vivo and in vitro samples, cells were permeabilised with 0.2% Triton X-100 for 10 min and blocked for 1 h in blocking buffer (3% goat serum and 3% BSA in PBS) before incubation with either pneumococcal antisera Pool Q (for 6B or 23F detection, SSI Diagnostica), or pneumococcal antisera Type 4 (for TIGR4 and dPly detection, SSI Diagnostica) to detect pneumococci, and JAM-A, Claudin-4 or β-catenin primary antibodies to detect cellular junctional proteins (see Supplementary Tables) for 1 h and then secondary and/or conjugated antibodies for 45 min. DAPI solution was added for 5 min. After washing, the stained samples were mounted using Aqua PolyMount (VWR International) with a coverslip onto a microslide. The entire cytospin for each sample was manually viewed by microscopy for detection of pneumococci. Multiple fields of view (>3) were imaged for each transwell insert, for each condition. Images were captured using either an inverted LSM 700, LSM 880 and analysed using the Zeiss LSM Image Browser, or captured using a TissueFAXS Zeiss Confocal Microscope and analysed with TissueGnostics software. Z stacks were recorded at 1 μm intervals at either ×40 oil or ×63 oil objectives. For the in vivo samples, the operator (CMW) was blinded to the colonisation status of the volunteer at the time of sampling.

**Electron microscopy**. Detroit 562 cells cultured on either transwells (scanning EM) or coverslips (transmission EM) were fixed with 2% paraformaldehyde and 1.5% glutaraldehyde in 0.1 M cacodylate buffer and post-fixed in 1% $OsO_4$/1.5% $K_4Fe(CN)_6$ in 0.1 M cacodylate buffer pH 7.3. Samples were dehydrated in graded ethanol–water series and infiltrated with Agar 100 resin mix. Ultra-thin sections were cut at 70–80 nm using a diamond knife on a Reichert ultracut microtome. Sections were collected on 300 mesh copper grids and stained with lead citrate. Samples were viewed in a Joel 1010 transition electron microscope and images recorded using a Gatan Orius camera. For scanning electron microscopy, cells were fixed in 2% paraformaldehyde and 1.5% glutaraldehyde in 0.1 M sodium cacodylate buffer pH 7.3 and incubated for 24 h at 3 °C. Samples are washed twice in 0.1 M sodium cacodylate for 30 min. The samples were post-fixed in 1% $OsO_4$/1.5% $K_4Fe$ $(CN)_6$ in 0.1 M cacodylate buffer at 3 °C for 1.5 h. Samples were washed in 0.1 M cacodylate buffer, rinsed in distilled water and dehydrated in a graded

ethanol–water series to 100% ethanol (25% ethanol 5 min; 50% ethanol 5 min; 70% ethanol 5 min; 90% ethanol 5 min and finally 100% ethanol (Annular and dry) 5 min, four times). Samples were critical point dried using $CO_2$ and mounted on aluminium stubs using sticky carbon taps. The mounted samples were coated with a thin layer of ~10-nm thick Au using a sputter coater and viewed and images recorded with a FEI.

For transmission electron microscopy, cells were fixed in 2% paraformaldehyde and 1.5% glutaraldehyde in 0.1 M sodium cacodylate buffer pH 7.3 and incubated for 24 h at 3 °C. Samples are washed twice in 0.1 M sodium cacodylate for 30 min and post-fixed in 1% $OsO_4$/1.5% $K_4Fe(CN)_6$ in 0.1 M cacodylate buffer pH 7.3 for 1 h at 3 °C. Samples are washed in distilled water for 5 min. The cells were dehydrated in a graded ethanol–water series and infiltrated with Agar 100 epoxy resin by the following steps: 25% ethanol 5 min; 50% ethanol 5 min; 70% ethanol 5 min; 90% ethanol 5 min; 100% ethanol (dry) 5 min, four times. This was followed by incubation in propylene oxide for 5 min; 2:1 propylene oxide:resin for 1 h; 1:1 propylene oxide 1 parts resin mix × 1 h; 1:2 propylene oxide:resin for 1 h; 100% resin for 4 h; 100% resin for 16 h. The epoxy resin was a medium hardness mix and the Agar resin consisted of Agar 100 (12 gm), DDSA (8 gm), MNA (5 gm) and BDMA (0.65 ml). The samples were hardened ready to section by placing the coverslip cell-side down onto a resin-filled beam capsule and hardened at 60 °C for 48 h. The coverslip was removed, and a representative area was selected. Ultra-thin sections were cut at 70–80 nm using a diamond knife on a Reichert ultra-cut S microtome. Sections were collected on copper grids and stained with lead citrate. Samples were viewed with a Joel 1010 transition electron microscope and the images recorded using a Gatan Orius CCD camera.

**Flow cytometry**. For the in vivo analysis, cells from two curette samples were incubated in cold PBS++ (PBS supplemented with 5 mM EDTA and 0.5% FCS) were dislodged by pipetting and centrifuged at 440 g for 5 mins at 4 °C. Supernatant was removed, and cells resuspended in 25 μl of PBS++ with Live/Dead™ Fixable Violet Dead Cell Stain (ThermoFisher). After 15 min incubation on ice, an antibody cocktail to stain for epithelial surface marker expression (see Supplementary Information) was added and incubated for another 15 min. Samples were vortexed, resuspended in 3.5 ml of PBS++ and filtered over a pre-wetted 70 μm filter. Samples were transferred to a 5 ml FACS tube, centrifuged and resuspended in 200 μl Cell Fix (BD Biosciences) and acquired on LSRII Flow Cytometer (BD Biosciences). Compensation was run and applied for each experimental replicate and voltages consistent throughout. Isotype controls (Biolegend) and FL-1 controls were also run for each antibody. Analyses of data was performed using FlowJo LLC version 10 software. Data were performed on the gated epithelial cell population and only samples containing 500 or more cells are reported.

For the in vitro analysis, confluent monolayers of Detroit 562 cells on 6-well plates were incubated with *S. pneumoniae* for 6 h in 1% FCS phenol free alpha MEM (base media, Life Technologies). Cells were washed three times in PBS and gently lifted from the plate using a cell scraper in 300 μl of base media supplemented with 1 mM EDTA. Samples were transferred to 5 ml FACS tubes and placed on ice for the duration of the protocol. Each cell sample was incubated with an antibody cocktail for epithelial surface marker expression (see Supplemental Information) and Live/Dead™ Fixable Zombie Blue UV Cell Stain (Biolegend) for 30 min before rinsing in 1 ml base media and centrifuging at $300 \times g$ for 5 min at 4 °C. Cells were fixed in 600 μl of 4% PFA and acquired on an LSR II Flow Cytometer (BD Biosciences). Compensation was run and applied for each experimental replicate and voltages consistent throughout. Isotype controls (BD Biosciences), FL-1 and single stains were also run for each experiment. Samples were acquired until 300,000 events had been collected. Analyses were performed using FlowJo LLC version 10 software.

**ELISAs**. Supernatant from Detroit 562 cells that had been incubated with *S. pneumoniae* for 6 h, was collected for cytokine analysis. IL-6, IL-8 and ICAM-1 DuoSet® ELISA kits were purchased from R&D Systems and protocol followed according to manufacturers' instructions.

**RNA samples and sequencing (RNASeq)**. Mucosal curettage samples and epithelial cell cultures (incubated with or without *S. pneumoniae* for 3 h) were collected in RNALater (ThermoFisher) and stored at −80 °C. RNA extraction was performed using the RNEasy micro kit (Qiagen) and genomic DNA was removed with on column DNA digestion or with the Turbo DNA-free Kit (Qiagen). Extracted RNA quality was assessed and quantified using a BioAnalyser (Agilent 2100). For mucosal curettage samples, library preparation and RNA sequencing (Illumina Hiseq 4000, 15 lanes in total with seven samples per lane, 100 bp and 75 bp paired-end reads) were performed at the Beijing Genome Institute (China) or the Wellcome Sanger Institute (UK), respectively. For the Detroit 562 samples, library preparation was performed using the Kappa Hyperprep kit (Roche Diagnostics) and sequencing was undertaken by the Pathogens Genomic Unit (UCL) on the Illumina Nextseq using the Nextseq 500/550 High Output 75 cycle kit (Illumina) according to manufacturers' instructions, giving 15–20 million 41 bp paired-end reads per sample.

All data processing and analysis were conducted in R, language and environment for statistical computing (https://www.R-project.org). Paired-end

reads were mapped to the Ensembl human transcriptome reference sequence (homo sapiens GRCh38). Mapping and generation of read counts per transcript were performed using Kallisto[63], based on pseudoalignment. R/Bioconductor package Tximport was used to import the mapped counts data and summarise the transcripts-level data into gene-level data[64]. DESeq2 and SARTools packages[65] were used for differential gene expression analysis, using a false discovery rate (FDR) <0.05, following normalisation with a negative binomial generalised linear model. In addition, transcript abundance for protein coding genes was expressed as $\log_2$-transformed transcripts per million (TPM) by normalising the read counts to gene length and then read depth. Transcriptomes from nasal samples ranged from 16 to 66 million reads per sample and in principle component analysis, revealed a batch effect that was corrected by the Combat function in the surrogate variable analysis (SVA) R/Bioconductor package[66].

Identification of interacting networks and pathway-enrichment analysis was performed using XGR R package[67]. Interacting genes (supported by at least one experimental source of evidence) were identified from the Pathway Commons database for directed interactions. These were then subjected to reactome pathway-enrichment analysis with FDR <0.05. K-means clustering of Jaccard indices to quantify similarity between the composition of genes mapping to each pathway was used to identify 20 groups of pathways, from which the pathway consisting of the largest total number of genes was selected to provide a representative annotation for each group. Transcription factor binding site enrichment analysis was performed using the human single site analysis function in oPOSSUM[68]. The expression of gene modules in the nasal transcriptomes was represented by the mean $\log_2$ TPM value of the genes in each module.

**Statistics**. All experiments were conducted with replicates in three or more independent experiments unless stated otherwise. Error bars represent SEM unless stated otherwise. GraphPad Prism Version 10 was used to perform two-tailed parametric (*t*-tests or ANOVA) or non-parametric (Mann–Whitney or Kruskal–Wallis tests) analysis, which was based on the Shapiro–Wilk normality test. P values <0.05 were considered significant and are derived from non-parametric statistical tests unless stated otherwise.

**Reporting summary**. Further information on research design is available in the Nature Research Reporting Summary linked to this article.

## Data availability
RNAseq data from the EHPC are accessible through GEO accession code GSE124949. RNAseq data from the Detroit 562 cell infections are accessible through the ArrayExpress database at EMBL-EBI under accession number E-MTAB-7841. Data from differential gene expression from upregulated genes following infection, Reactome pathways and transcription factor analysis can be found in Supplementary Data 1–4. Source data for figures are provided as a Source Data file.

## Code availability
Codes used to process the RNAseq data can be found at https://github.com/cristina86cristina.

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

## Acknowledgements

This study was funded by the Wellcome Trust (Grant 106846/Z/15/Z). CW and RSH are supported by the NIHR Global Health Research Unit on Mucosal Pathogens (Grant 16/136/46) using UK aid from the UK Government. The views expressed in this publication are those of the author(s) and not necessarily those of the NIHR or the Department of Health and Social Care. D.M.F. is supported by the Medical Research Council (grant MR/M011569/1), Bill and Melinda Gates Foundation (grant OPP1117728) and the National Institute for Health Research (NIHR) Local Comprehensive Research Network. The authors wish to thank the EHPC Clinical Team at LSTM and all the volunteers. Confocal imaging facilities at LSTM were funded by a Wellcome Trust Multi-User Equipment Grant (104936/Z/14/Z). Flow cytometric acquisition at LSTM was funded by a Wellcome Trust Multi-User Equipment Grant (104936/Z/14/Z). LytA PCR was performed by Prof. D Bogaert, University of Edinburgh, UK. For in vitro data, RNAseq library preparation was undertaken at UCL through the UCL/UCLH Biomedical Research Centre and MRC funded Pathogen Genomics Unit (G0900950). For the in vivo data, RNAseq library preparation was performed and by Carl Anderson at the Wellcome Sanger Institute, Hinxton. E.M. was performed by Dr. Mark Turmaine in the Biosciences EM facility at UCL. M.N. is supported by the Wellcome Trust. M.N and J.S.B are supported by the NIHR Biomedical Research Centre Funding to University College Hospitals NHS Foundation Trust and University College London.

## Author contributions

C.M.W., S.P.J., D.M.F. and R.S.H. conceived and designed the study. C.M.W., S.P., S.P.J., J.R., E.N. and C.S. acquired the data. C.M.W., C.V., S.P.J., M.N., J.S.B., D.M.F. and R.S.H. analysed and interpreted the data. C.M.W. wrote the first draft of the paper. C.M.W., C.V., S.P., S.P.J., J.R., E.N., C.S., M.N., J.S.B., D.M.F. and R.S.H. commented on and approved the paper.

## Additional information

**Competing interests:** The authors declare no competing interests.

