## [Peer Review File · Nature Communications]

Reviewers' comments:

Reviewer #2 (Remarks to the Author):

The authors' make a compelling argument using biopsies from colonized versus non colonized volunteers exposed to pneumococcal serotype 6B; as well as cell culture studies using an additional invasive serotype 4 strain and its pneumolysin deficient mutant, as well as another colonizing 23F strain that micro-invasion of the epithelium is a feature of colonization.

As the hemolytic effects of the pneumolysin did not predict micro-invasion, what properties of the pneumolysin do the authors' suggest plays a role in this phenomenon?

The authors' speculate on the role of viruses in colonization - were viruses sought by PCR in their colonization studies and did they exclude a role for viruses in prediction of colonization in these patients - this group has looked at viral factors associated with colonization and do they inform the process of micro-invasion?

The authors' speculate that serotype 4 invasiveness can be measured by the amount of micro-invasion in their in vitro assay - it is interesting to think that this assay be used to assess the invasiveness of pneumococcal types - and also the invasiveness of different genetic lineages within serotype.

As serum and local antibody responses were presumably measured - did they correlate with the degree of micro-invasion or any of the other markers tested?

Reviewer #3 (Remarks to the Author):

The work by Weight et al. is interesting and has potential paradigm shifting findings on the maintenance and clearance of pneumococcal carriage. The manuscript provides evidences to support the important role of epithelial sensing of pneumococci in the outcome of carriage using in vivo and in vitro experimental models. The authors also define a novel mechanism of pneumococcal invasion, termed micro-invasion, which allows migration across epithelial barrier without overt disease. They show that pneumococcal engagement with epithelial cells does not always lead to inflammation but that micro-invasion is the main process that elicits innate immunity/inflammation. In conclusion, the authors suggest that micro-invasion induced innate immunity/inflammation promotes clearance and may trigger transmission of pneumococci.

Major comments:

General

1. The authors relate the importance of micro-invasion to transmission yet they do not provide data supporting the suggestion. May the authors include a paragraph in the introduction section on how pneumococcal transmission happens. They should also include in the discussion section more details on how innate immunity/inflammation can enhance transmission of pneumococci.
2. The flow of the results section is less coherent and in some cases a bit confusing. The authors should consider revising the results by including all results for one concept under one heading. They can first show the in vivo findings and support them with in vitro finding for each concept. For example, the findings under the first heading and third heading under the results should one, with the first section of heading dealing with in vivo data and second section in vitro data.
3. The authors should provide more detail on the sample sizes per experiment instead of stating $n > 2$ or $n > 3$ or $n > 20$.
4. The authors should consider revising the title of the manuscript to reflect their major findings

Specific

1. Line 118, Figure 1e. What time point was micro-colony formation observed in the in vivo model? Do the authors have data on how long does it take to observe micro-colony formation in the in vitro model?

2. Line 125-141: Why were these specific markers chosen? Are these the main classic markers for epithelial activation?
5. Line 267-268: The authors should provide more data supporting their results. Supplementary Figure 9 does not provide enough detail. Where are stated genes and innate molecules coming from? May they also provide more detail on how the data points towards innate activation?
6. Line 284: The data in Figure 1b shows decrease in CFUs at day 9, how do the authors reconcile this with their statement that "prominence of responses at day 9 may reflect time-dependent proliferation of the 6B bacterial inoculum"?
7. Line 336: How would innate immune/inflammatory responses promote survival through transmission?
8. Line 355-358: The conclusions need to be rephrased to align with the actual findings of the study. The authors have not provided data on transmission, hence it would be premature to be base the conclusion on the balance between clearance and transmission. If the authors do not have transmission data then they should state this as one of the limitations of the study.
9. Line 358-360: These statements need to be combined into one sentence to improve clarity of the message.

Minor comments:

1. Line 62: The authors should include the role of anti-polysaccharide antibodies in natural acquired immunity
2. Line 78-80: The authors have should include a summary of the major findings from the EHPC model, to help contextualise the current study.
3. Line 107-110: This should be in the discussion section
4. Line 157: The authors need to provide more detail on their rationale for choosing some of the markers, including B catenin and Claudin 4
5. Line 196: What about dPLY?
6. Line 252-254: The authors should consider revisiting the sentence as it is not clear.
7. Figure 7b: Use better colours in the graphs to distinguish the two groups
8. Line 281-283: Sentence needs rephrasing
9. Line 330-334: Only the relevance of KLF4 is discussed, what about SP1?
10. Line 344-346: How is pneumococcal DNA release related to absence of pneumolysin?
11. Line 351-352: The relevance of this statement requires a bit more explanation.
12. Line 659-661: Is there any reason why data from the other cytokines are not shown?

Figure legends

Figure 1c: How were the numbers calculated? What do the horizontal lines on the bar graphs represent? Why only 8 volunteers?

Figure 1d: What do the numbers 1-4 represent?

Figure ei: Which day was this?

Figure eii: Is the Red representing JAM-A or WGA?

Figure 2: Why are the number of participants changing over time? Please include the p value

Figure 3e-h: Which strain was used?

Figure 4c: TEER should be stated in full i.e. Trans-Epithelial Electrical Resistance

Figure 5a-b: The figure needs to be reworked to clearly show which comparisons are being made, drawing lines linking the groups from which the p value is derived would be helpful.

Figure 5c: Are all these comparison against non-infected samples? See comment above

Figure 7: The number of samples is greater than the 13 stated in Table 1. Any explanation on the extra samples?

Figure 8: Line 821-823: The two sentences needs to be rephrased to reflect the potential changes to the manuscript.

Reviewer #4 (Remarks to the Author):

1. BASIC REPORTING

To analyze the mechanism of colonization the authors combined an EHPC model and in vitro human cell culture systems. While in the EHPC model only strain 6B was used, different serotypes and a ply-negative strain of TIGR4 were used in the in vitro assays. The authors indicated that pneumococcal colonization leads to epithelial adherence (a result confirming previous studies), but also to microcolony formation and migration across the barrier. Because of the use of different pneumococcal strain, this studies indicates and confirms that the efficiency of these processes are strain dependent. The association of pneumococci with junctional proteins has been demonstrated and suggested to promote micro-invasion. The authors made the conclusion, via analysis of the host response by transcriptomics and measuring of cytokine profiles, that micro-invasion and pneumolysin trigger the epithelial sensing process. As a consequence the micro-invasion may influence outcome of colonization or even transmission.

In conclusion, this excellent study confirms earlier in vitro and mouse in vivo studies, but by engagement of the sophisticated EHPC model this study points further to important new aspects of colonization and epithelial sensing.

Despite this study is excellent and most of the assays are perfectly designed, there are a few aspects that should be considered in a revised version.

2. EXPERIMENTAL DESIGN

- Most of the experiments are excellently described and the information needed to perform the experiments are provided.
- There is, however, one aspect which has not been mentioned throughout the manuscript. Despite showing fluorescently labeled pneumococci it is not mentioned whether these are GFP-expressing pneumococci – which is at least probably not the case in the EHPC model (with the exception once in the SI) or whether the authors have conducted an immunofluorescent (IF) staining of the bacteria (see also minor comments). If the IF staining has been perform the next question arises how the cells were treated to enable discrimination between extracellular and intracellular bacteria

3. VALIDITY OF THE FINDINGS

- The statistical analysis is approved

4. General comments

Although the manuscript and the data are overall convincing, the reviewer has one important concern regarding the conclusions of pneumococcal invasion and transmigration. The authors claim that the visualization of pneumococci on epithelial cells (EHPC model or immortalized epithelial cells, is indicative of cell-attached bacteria (which is fine), but also of intracellular and transmigrated (or in the process of transmigration) bacteria. The method of choice is the immunofluorescence microscopy (IFM). Although the IFM is extremely useful the data may suggest that there are intracellular bacteria. However, only 0.005% of the adherent pneumococci are internalized (calculation according to the gentamicin assay). In total at max 5000 (dply-TIGR4 in Fig. 3) are recovered from the intracellular compartment of cells. Considering the methodology of staining (?) and images provided in Fig.1, Fig. 3e this conclusion is not justified. As indicated by the gentamicin assay higher levels of adherence results in higher numbers of intracellular bacteria. Nevertheless, the final evidence of intracellular pneumococci can only be achieved by electron microscopy. Other (in vitro) studies have indicated that uptake of pneumococci happens, but still this is a low proportion of bacteria that are taken up by probably different mechanisms. Although the reviewer likes the term micro-invasion, this term has to be defined precisely for the ongoing processes on the epithelial cell during pneumococcal colonization. Maybe the pneumococcus is tightly attached and only partially taken up by the cell via inducing signal cascades. Similar, the authors use the term microcolony formation. It's obvious from the

immunofluorescence images that not all cells are colonized by pneumococci. Why this is the case has not been discussed at all and is probably out of the scope of this study. However, the microcolonies might also be chains of pneumococci (as it looks in Fig. 1 ei) and this can only be answered perfectly by using electron microscopy (EM). In the view of the reviewer formations of microcolonies is different from the enhanced numbers of attached pneumococci, which can probably be induced by the formation of chains instead of diplococci (here the size of bacteria may determine the induced host response as well). Again, EM will provide more details.

Minor comments:

1. Page 5, line 122: ... is suggestive of an active process. Please explain which active process is meant and who triggers this active process; is the pneumococcus the active part or is the active process triggered by the induced host response
2. Throughout fluorescence microscopy visualization of pneumococci it is not mentioned how pneumococci were labeled. Assuming that in the EHPC model no GFP-pneumococci could be used pneumococci seem to be labeled by immunofluorescence staining. Please provide details. This is in particular of interest in the context of host cell attached and intracellular pneumococci. Without permeabilization of cells and double immunofluorescence staining a discrimination between extra- and intracellular bacteria is not valid.
3. Is this correct that pneumococci have been killed by using 100 µg/ml gentamicin without addition of penicillin?
4. Although several studies still mention that lipoteichoic acids trigger TLR signaling this is not correct as other studies have clearly indicated that this occurs via lipopeptides. Please see and change this part on page 3, line 73

Supplementary material:

5. Reactome: the link to the excel spreadsheets should finally be included

Authors responses to Reviewers' comments on “Epithelial control of colonisation by *Streptococcus pneumoniae* at the human mucosal surface”
NCOMMS-18-24929-T.

Reviewer #2 (Remarks to the Author):

As the hemolytic effects of the pneumolysin did not predict micro-invasion, what properties of the pneumolysin do the authors' suggest plays a role in this phenomenon?

Response: We have shown that micro-invasion is not pneumolysin dependent. However, comparison of the TIGR4 pneumolysin mutant and the wild-type strain suggests that pneumolysin enhances an innate-epithelial response, as shown by epithelial CD107a expression and RNAseq. Upon internalisation of the pneumococcus by neutrophils, pneumolysin has been shown to induce ROS following bacterial autolysis which leads to cellular activation (Martner, Infection and Immunity 2008). We speculate that pneumolysin released intracellularly may signal directly in epithelial cells or through host cell pore-formation leading to entry of other PAMPs.

While the elucidation of these pathways is beyond the scope of the present manuscript, this has now been clarified in the discussion (lines 388-391).

The authors' speculate on the role of viruses in colonization - were viruses sought by PCR in their colonization studies and did they exclude a role for viruses in prediction of colonization in these patients - this group has looked at viral factors associated with colonization and do they inform the process of micro-invasion?

Response: The reviewer is correct that in the EHPC model, asymptomatic viral infection is associated with increased rates of colonisation (Glennie, Mucosal Immunology, 2015). Indeed, immunisation with live-attenuated influenza vaccine (LAIV) is associated with increased pneumococcal carriage density (Rylance, Biorxiv 2018). In the current study, EHPC model volunteers who had coryzal or flu-like symptoms prior to challenge were excluded. All volunteers were screened at baseline for 20 viruses by PCR. One asymptomatic volunteer (volunteer 1) was positive for enterovirus. This has now been clarified in the introduction (lines 60-62) and methods (lines 645-648).

The authors' speculate that serotype 4 invasiveness can be measured by the amount of micro-invasion in their in vitro assay - it is interesting to think that this assay be used to assess the invasiveness of pneumococcal types - and also the invasiveness of different genetic lineages within serotype.

Response: We agree with the reviewer that our data highlights an interesting area in relation to invasiveness. However, we have been careful not to imply that micro-invasiveness equates with potential to cause disease as this may not be the case.

As demonstrated by the EHPC model, in adults who are presumed to have some anti-pneumococcal immunity, microinvasion may occur without disease. We have modified the discussion to clarify this point (lines 356-358).

As serum and local antibody responses were presumably measured - did they correlate with the degree of micro-invasion or any of the other markers tested?

Response: This question highlights the complexity of naturally acquired immunity to the pneumococcus. None of the volunteers had received pneumococcal vaccine. Most adults are considered to have anti-pneumococcal immunity which confers some protection. There is considerable controversy as to what elements of natural immunity are protective, but the consensus is that anti-protein T cell and antibody mediated immunity rather than anti-capsular immunity are most important (Jochems et al). Previous EHPC studies in the same population have demonstrated that all healthy adult volunteers have detectable anti-pneumococcal immunity (Wright and Ferreira et al, PLoS Pathogens 2012). Antibody, particularly at the mucosal surface, was not systematically measured in this cohort of volunteers. Additionally, we would have been underpowered to elicit any subtle differences, if present. This is now addressed in the introduction (lines 64-68) and methods section (lines 642-645). We have not pursued this point further in the discussion as this is beyond the scope of this manuscript.

Reviewer #3 (Remarks to the Author):

Major comments:

General

1. The authors relate the importance of micro-invasion to transmission yet they do not provide data supporting the suggestion. May the authors include a paragraph in the introduction section on how pneumococcal transmission happens. They should also include in the discussion section more details on how innate immunity/inflammation can enhance transmission of pneumococci.

Response: A sentence in the introduction has been added (lines 60-62) and in the discussion (lines 373-376).

2. The flow of the results section is less coherent and in some cases a bit confusing. The authors should consider revising the results by including all results for one concept under one heading. They can first show the in vivo findings and support them with in vitro finding for each concept. For example, the findings under the first heading and third heading under the results should one, with the first section of heading dealing with in vivo data and second section in vitro data.

Response: While the authors appreciate that the narrative could be separated into concepts, as described by the reviewer, we think that this could be more confusing to

the reader, breaking up the chain of evidence from *in vivo* to *in vitro* and then back to *in vivo*. We prefer to first focus on the *in vivo* EHPC model to characterise epithelial colonisation and then evaluate the mechanisms in the epithelial model in detail. To demonstrate the *in vivo* relevance of the RNAseq data generated from the Detroit 562 cells, we have generated modules to probe the EHPC transcriptomic response.

We have modified the manuscript to strengthen our narrative.

If the editor should prefer the narrative to be redesigned according to the reviewer's suggestions, we will further modify the manuscript.

3. The authors should provide more detail on the sample sizes per experiment instead of stating n>2 or n>3 or n>20.

Response: These have now been included in the figure legends.

4. The authors should consider revising the title of the manuscript to reflect their major findings

Response: The current title is: "Epithelial control of colonisation by *Streptococcus pneumoniae* at the human mucosal surface". In light of the reviewers' suggestion, we have changed the title to "Epithelial micro-invasion by *Streptococcus pneumoniae* induces epithelial-derived innate immunity during colonisation at the human mucosal surface".

Specific

1. Line 118, Figure 1e. What time point was micro-colony formation observed in the *in vivo* model? Do the authors have data on how long does it take to observe micro-colony formation in the *in vitro* model?

Response: Micro-colony formation was observed on day 6 and day 9 post inoculation in the EHPC model. *In vitro*, micro-colony formation was readily apparent at three hours post infection, but not observed at shorter time periods. We postulate that micro-colony formation represents replication at the epithelial surface but in the absence of live imaging, this is not possible to substantiate.

We have modified the results to clarify this point at line 126.

2. Line 125-141: Why were these specific markers chosen? Are these the main classic markers for epithelial activation?

Response: CD107a, IL-22Ra, HLA-DR, CD40 and CD54 are known to be expressed in airway epithelium, are upregulated in response to common bacterial stimuli and are involved in immune cell recruitment.

This has been further clarified in the results section in lines 147-150.

5. Line 267-268: The authors should provide more data supporting their results. Supplementary Figure 9 does not provide enough detail. Where are stated genes and innate molecules coming from? May they also provide more detail on how the data points towards innate activation?

Response: We now include the excel spreadsheets containing a list of genes that are enriched from the *in vivo* and *in vitro* RNAseq data, as part of the supplementary information. In relation to the *in vivo* transcriptome, this component of the analysis was not cell-type specific, hence the use of the *in vitro*-derived modules subsequently to identify the epithelial response.

We have updated the paragraph for clarity and removed supplementary figure 9 from the manuscript.

6. Line 284: The data in Figure 1b shows decrease in CFUs at day 9, how do the authors reconcile this with their statement that “prominence of responses at day 9 may reflect time-dependent proliferation of the 6B bacterial inoculum”?

Response: The intention of this statement was to summarise the process of initial epithelial contact, bacterial proliferation and micro-colony formation, micro-invasion and the triggering of an innate inflammatory response. This inflammatory response was most prominent around day 9 when as stated by the reviewer, clearance has started to occur.

This has been further clarified in the text in line 320.

7. Line 336: How would innate immune/inflammatory responses promote survival through transmission?

Response: It has been postulated that respiratory droplet spread of *S. pneumoniae* is promoted by mucosal inflammation and increased mucus production (Siegel and Weiser). Weiser and colleagues have shown that pneumolysin mediated-inflammation increases bacterial shedding in a murine model (Zafar et al).

This has been further clarified in the discussion between lines 372-376.

8. Line 355-358: The conclusions need to be rephrased to align with the actual findings of the study. The authors have not provided data on transmission, hence it would be premature to be base the conclusion on the balance between clearance and transmission. If the authors do not have transmission data then they should state this as one of the limitations of the study.

Response: A statement on line 404-405 has been included in the limitations paragraph. Lines 407-418 have been rephrased as suggested.

9. Line 358-360: These statements need to be combined into one sentence to improve clarity of the message.

Response: These sentences have been combined as suggested (lines 415-418).

Minor comments:

1. Line 62: The authors should include the role of anti-polysaccharide antibodies in natural acquired immunity

Response: This has now been incorporated into the sentence (lines 64-66).

2. Line 78-80: The authors have should include a summary of the major findings from the EHPC model, to help contextualise the current study.

Response: Lines 88-89 have been added to the manuscript to incorporate the reviewer's suggestion.

3. Line 107-110: This should be in the discussion section

Response: The purpose of this sentence (line 113-119) is as a clarification. We feel that it would be out of place in the discussion. Should the editor prefer us to move the sentence to the discussion, we will do so but this will increase the size of the manuscript.

4. Line 157: The authors need to provide more detail on their rationale for choosing some of the markers, including B catenin and Claudin 4

Response: An explanation has been incorporated into the text (lines 181-184).

5. Line 196: What about dPLY?

Response: (lines 225-226) Although there was a trend towards a drop in resistance, the difference between non-infected cells and those infected with the dPLY-TIGR4 strain were not statistically significant.

6. Line 252-254: The authors should consider revisiting the sentence as it is not clear.

Response: The sentence has been revised in lines 285-287.

7. Figure 7b: Use better colours in the graphs to distinguish the two groups

Response: We have intensified the colour separation of black and grey.

8. Line 281-283: Sentence needs rephrasing

Response: The sentence has been revised in lines 318-321.

9. Line 330-334: Only the relevance of KLF4 is discussed, what about SP1?

Response: Discussion of the relevance of SP1 in the data set would be purely speculative as we could find no relation to pneumococcal infection in the literature. Therefore, SP1 has been removed from line 379.

10. Line 344-346: How is pneumococcal DNA release related to absence of pneumolysin?

Response: The detection of components of the Type 1 interferon signalling pathway in the dPly-TIGR4 strain suggests that DNA release is not entirely dependent on the presence of pneumolysin. Pneumococcal DNA release is known to also be initiated by the activity of autolysin and bacterial death. We and others postulated that intracellular bacterial autolysis releases pneumococcal DNA into the epithelial cytoplasm. This has been clarified in the text (lines 389-392).

11. Line 351-352: The relevance of this statement requires a bit more explanation.

Response: The statement has been revised in lines 405-406.

12. Line 659-661: Is there any reason why data from the other cytokines are not shown?

Response: Line 761. IL-1 β , TNF α and IFN γ were not expressed by the epithelial cells. They were included in the panel of cytokines to initially screen epithelial activation since they are associated with a proinflammatory phenotype following pneumococcal infection. This has been deleted from the text. We can include the data in the supplemental information at the editors' request.

Figure legends

Figure 1c: How were the numbers calculated? What do the horizontal lines on the bar graphs represent? Why only 8 volunteers?

Response:

The entire sample was visually screened, and every single bacterium found was imaged. All images were analysed for numbers of bacteria, performed by manual counts for the total; surface-associated, intracellular, and laterally located. For Figure 1c, the error bars are standard error across all time points for all volunteers who were carriage positive. Out of the 13 volunteers, 8 were positive carriers. The graph is based on their bacterial counts.

This has been clarified in the text on line 843.

Figure 1d: What do the numbers 1-4 represent?

Response: The numbers represent the top and bottom of the sample depth. Each image was taken using 1µm slice, so the total depth that is shown is 4µm.

This has been clarified in the figure legend on line 846.

Figure ei: Which day was this?

Response: This was day 6 post inoculation.

This has been clarified in the figure legend on line 847.

Figure eii: Is the Red representing JAM-A or WGA?

Response: Figure 1e is representing WGA.

This has been clarified in the figure legend on line 847.

Figure 2: Why are the number of participants changing over time? Please include the p value

Response: Lines 860-861. The samples used from this EHPC cohort were collected during a clinical trial. Volunteers who received live attenuated influenza vaccine two days post pneumococcal challenge were therefore subsequently excluded from further analysis, only contributing baseline and day 2 results in the dataset presented. In addition, as mentioned in the methods, samples containing less than 500 epithelial cells were excluded from the final flow cytometry analyses. Together, these explain the differences in sample sizes (summarised in Supplemental Figure 2b).

P values have now been included in the legend.

Figure 3e-h: Which strain was used?

Response: The figure legends have now been amended in lines 874-877:

Figure 3f top panel – 6B. Bottom panel – TIGR4.

Figure 3g – top left and middle – TIGR4, right panel – 6B

Figure 3h – 23F

Figure 4c: TEER should be stated in full i.e. Trans-Epithelial Electrical Resistance

Response: The text has been altered for lines 224 and 893, Figure 4c and in the supplementary information and figures.

Figure 5a-b: The figure needs to be reworked to clearly show which

comparisons are being made, drawing lines linking the groups from which the p value is derived would be helpful.

Figure 5c. Are all these comparison against non-infected samples? See comment above

Response: The comparison has been clarified in the figure.

Figure 7: The number of samples is greater than the 13 stated in Table 1. Any explanation on the extra samples?

Response: Due to limitations in sample availability, the RNAseq data and the confocal microscopy & flow cytometry data could not be collected from all the same volunteers. Therefore, the number of samples are not the same.

Figure 8: Line 821-823: The two sentences needs to be rephrased to reflect the potential changes to the manuscript.

Response: (Lines 940-949). The figure title and paragraph has been rephrased as suggested.

Reviewer #4 (Remarks to the Author):

2. EXPERIMENTAL DESIGN

• There is, however, one aspect which has not been mentioned throughout the manuscript. Despite showing fluorescently labelled pneumococci it is not mentioned whether these are GFP-expressing pneumococci – which is at least probably not the case in the EHPC model (with the exception once in the SI) or whether the authors have conducted an immunofluorescent (IF) staining of the bacteria (see also minor comments). If the IF staining has been perform the next question arises how the cells were treated to enable discrimination between extracellular and intracellular bacteria

Response: In all the data presented in the main manuscript, pneumococci were identified using either pneumococcal antisera Pool Q (for 6B and 23F detection), or pneumococcal antisera Type 4 (for TIGR4 and dPly-TIGR4 detection) followed by a fluorescent secondary antibody. This has been clarified in the methods (Ines 706-708) and supplementary figures.

4. General comments

Although the manuscript and the data are overall convincing, the reviewer has

one important concern regarding the conclusions of pneumococcal invasion and transmigration. The authors claim that the visualization of pneumococci on epithelial cells (EHPC model or immortalized epithelial cells, is indicative of cell-attached bacteria (which is fine), but also of intracellular and transmigrated (or in the process of transmigration) bacteria. The method of choice is the immunofluorescence microscopy (IFM). Although the IFM is extremely useful the data may suggest that there are intracellular bacteria. However, only 0.005% of the adherent pneumococci are internalized (calculation according to the gentamicin assay). In total at max 5000 (dply-TIGR4 in Fig. 3) are recovered from the intracellular compartment of cells. Considering the methodology of staining (?) and images provided in Fig.1, Fig. 3e this conclusion is not justified. As indicated by the gentamicin assay higher levels of adherence results in higher numbers of intracellular bacteria. Nevertheless, the final evidence of intracellular pneumococci can only be achieved by electron microscopy. Other (in vitro) studies have indicated that uptake of pneumococci happens, but still this is a low proportion of bacteria that are taken up by probably different mechanisms. Although the reviewer likes the term micro-invasion, this term has to be defined precisely for the ongoing processes on the epithelial cell during pneumococcal colonization. Maybe the pneumococcus is tightly attached and only partially taken up by the cell via inducing signal cascades. Similar, the authors use the term micro-colony formation. It's obvious from the immunofluorescence images that not all cells are colonized by pneumococci. Why this is the case has not been discussed at all and is probably out of the scope of this study. However, the micro-colonies might also be chains of pneumococci (as it looks in Fig. 1 ei) and this can only be answered perfectly by using electron microscopy (EM). In the view of the reviewer formations of micro-colonies is different from the enhanced numbers of attached pneumococci, which can probably be induced by the formation of chains instead of diplococci (here the size of bacteria may determine the induced host response as well). Again, EM will provide more details.

Response: As highlighted by the reviewer, only a small proportion of colonising pneumococci are internalised by the epithelial cells or undergo transmigration. As mentioned, we have shown this by a gentamicin protection assay and confocal microscopy, in which we have used staining with both capsular antibody (manuscript figures) and a stably expressed FAMSE fluorescent dye (supplementary Figure 3). These data are in line with the existing pneumococcal literature and for upper respiratory tract commensals in general.

Following the very helpful suggestion of the reviewer, we have now further interrogated these pneumococcal-epithelial interactions by electron microscopy using our epithelial cell culture model and nasal curette samples from healthy volunteers colonised with *S. pneumoniae*.

We have generated compelling novel scanning and transmission EM images from our *in vitro* model which show both surface-associated (in both chains and micro-colonies), intracellular/ intravacuolar and paracellular pneumococci (see new Figures 3i and 3j). We have also made a new observation of epithelial cell membrane folding during invasion of pneumococci into the cell which we have also included (Figure 3i and 3j).

It has proven more challenging to visualise pneumococcal-epithelial interactions in our *in vivo* human samples due to sample availability, the fragility of the tissue samples, the absence of an established fixation protocol and, the relatively low number of colonising bacteria and epithelial cells per sample. Indeed, compared with the confocal fluorescence microscopy, the ultra-resolution of EM is not an ideal tool for screening relatively large areas of tissue, and in depth. Nonetheless, following optimisation, we are able to report the architecture of the nasal cells from the curette biopsies visualised by TEM (Supplementary Figure 1bi and 1bii) which show microvilli on the epithelial surface (bi, MV), goblet cells (bi, G) and intact sheets of cells with tight junctions (bii). We also provide evidence suggestive of pneumococci in carriage positive individuals. We have found diplococci that may have been dislodged from the epithelial surface (Supplementary Figure 1biii and 1biv). While we cannot be certain that these are *S. pneumoniae* without immunostaining, they have the appropriate morphological features. As new techniques such as integrated fluorescence electron microscopy which allow the rapid localisation of events of interest become more established and more available, we plan to further pursue the ultrastructural relationship between the pneumococcus and the epithelium *in vivo*.

The methods, results and discussions have been modified accordingly to incorporate the new data.

We have defined the term micro-invasion in the introduction (lines 206-207) and discussion but have further included a definition clarity within lines 129-131 as the 'internalisation and/ or transmigration of pneumococci within the epithelium'. Micro-colony formation has been clarified on line 127-128 as 'two or more pneumococci co-located on an epithelial cell'.

Minor comments:

1. Page 5, line 122: ... is suggestive of an active process. Please explain which active process is meant and who triggers this active process; is the pneumococcus the active part or is the active process triggered by the induced host response

Response: Line 134-135 has been altered to address this point.

2. Throughout fluorescence microscopy visualization of pneumococci it is not mentioned how pneumococci were labeled. Assuming that in the EHPC model no GFP-pneumococci could be used pneumococci seem to be labeled by immunofluorescence staining. Please provide details. This is in particular of

interest in the context of host cell attached and intracellular pneumococci. Without permeabilization of cells and double immunofluorescence staining a discrimination between extra- and intracellular bacteria is not valid.

Response: Lines 706-709 have been expanded and relevant information included.

3. Is this correct that pneumococci have been killed by using 100 µg/ml gentamicin without addition of penicillin?

Response: This is correct. Gentamicin has been extensively used in the literature as an effective method of killing extracellular pneumococci in these protection assays. Penicillin was not used because of the concern that it may be absorbed into the cell, affecting viability of internalised bacteria.

4. Although several studies still mention that lipoteichoic acids trigger TLR signaling this is not correct as other studies have clearly indicated that this occurs via lipopeptides. Please see and change this part on page 3, line 73

Response: This has now been removed.

Supplementary material:

5. Reactome: the link to the excel spreadsheets should finally be included

Response: This has now been done.

REVIEWERS' COMMENTS:

Reviewer #2 (Remarks to the Author):

No further comments to the authors

Reviewer #3 (Remarks to the Author):

The authors have satisfactorily addressed all my comments